# A dynamic and thermodynamic analysis of the 11 December 2017 tornadic supercell in the Highveld of South Africa

Lesetja E. Lekoloane[1,2], Mary-Jane M. Bopape[1], Gift Rambuwani[1], Thando Ndarana[3], Stephanie Landman[1], Puseletso Mofokeng[1,4], Morne Gijben[1], and Ngwako Mohale[1]

[1]South African Weather Service, Pretoria, 0001, South Africa
[2]Global Change Institute, University of the Witwatersrand, Johannesburg, 2050, South Africa
[3]Department of Geography, Geoinformatics and Meteorology, University of Pretoria, Pretoria, 0001, South Africa
[4]School of Geography, Archaeology and Environmental Studies, University of the Witwatersrand, Johannesburg, 2050, South Africa

**Correspondence:** Lesetja Lekoloane (lesetja.lekoloane@weathersa.co.za)

**Abstract.** On 11 December 2017, a tornadic supercell initiated and moved through the northern Highveld region of South Africa for 7 hours. A tornado from this supercell led to extensive damage to infrastructure and caused injury to and displacement of over 1000 people in Vaal Marina, a town located in the extreme south of the Gauteng Province. In this study we conducted an analysis in order to understand the conditions that led to the severity of this supercell, including the formation

5 of a tornado. The dynamics and thermodynamics of two configurations of the Unified Model (UM) were also analysed to assess their performance in predicting this tornadic supercell. It was found that this supercell initiated as part of a cluster of multicellular thunderstorms over a dryline, with three ingredients being important in strengthening and maintaining it for 7 hours: significant surface to mid-level vertical shear, an abundance of low-level warm moisture influx from the tropics and Mozambique Channel, and the relatively dry mid-levels (which enhanced convective instability). It was also found that the 4.4

10 km grid spacing configuration of the model (SA4.4) performed better than the 1.5-km grid spacing version. SA1.5 underestimated the low-level warm moisture advection and convergence, and missed the storm initiation. SA4.4 captured the supercell; however, the mid-level vorticity was found to be one order of magnitude smaller than that of a typical mesocyclone. A grid length of 4.4km is too coarse to fully capture the details of a mesocyclone. The SA4.4 also underestimated the surface to mid-level wind shear and low-level horizontal mass and moisture flux convergence. One recommendation is that the

15 classical dynamic definition of mesocyclones be tested for subtropical regions, where vertical wind shears are typically weaker compared to the mid-latitudes. Future investigations will involve experimental research over the Highveld region of South Africa to understand mesoscale and local dynamics processes responsible for tornadogenesis in some severe storms. Such a study, to the best of our knowledge, has never been conducted.

## 1 Introduction

20 Human survival and development are directly and indirectly impacted by weather and climate extremes (Nuttall, 1998; IPCC, 2014). In their different spatial and temporal scales and reach, extreme weather events can result in prolonged negative socio-

economic impacts and affect different facets of the society, including ecosystems (Meehl et al., 2000). Examples of these extremes include powerful synoptic tropical systems such as Tropical Cyclones Eline (Reason and Keibel, 2004) and Idai (WMO, 2019) that occurred in the years 2000 and 2019, respectively. These severe storms affected millions of people in southern African mainland. Extreme weather events have been observed in other parts of the world, for instance the unusually extended tornado outbreak that occurred across the central and eastern United States during May 2003 and resulted in about $829 million of property damage  (Hamill et al., 2005). There is growing evidence suggesting that these extreme events are becoming more common across the globe and can be expected to increase in intensity and frequency due to climate change (e.g. Handmer et al., 2012; Melillo et al., 2014; EASAC, 2018).

There is no region on Earth immune to extreme or severe weather events, but the most vulnerable communities are especially those from developing countries (Mirza, 2003). For example, many developing countries' economies and livelihoods are highly depended on primary sectors of the economy (UN, 2018a) such as agriculture and fishing of which negative effects have already been experienced due to an increase in high impact weather events (Adger et al., 2003; Awojobi and Tetteh, 2017). Therefore, an understanding of these high impact weather events, and being able to predict them precisely, is of importance for sustainable development as well as mitigation and adaptation strategies.

One of the regions identified as highly vulnerable to climate change and its effects is sub-Saharan Africa (Boko et al., 2007). Most countries in this region of the world are lacking behind in the development scale and have many stressors including high impact weather events (UN, 2018b). Although no single weather event can be confidently attributed to climate change, an increasing number of studies are documenting a positive trend of severe weather events in southern Africa (Davis-Reddy and Vincent, 2017; Kruger and Nxumalo, 2017). These include an increase in the frequency of extreme rainfall events associated with land-falling tropical cyclones and severe thunderstorms, particularly in the eastern parts of southern Africa and south of South Africa (Malherbe et al., 2013; Engelbrecht et al., 2013).

A climatological analysis over South Africa indicates an increase in extreme rainfall events, especially during spring and summer months (Engelbrecht et al., 2013; DEA, 2016) mostly in the form of convective precipitation (Tyson and Preston-Whyte, 2000; Hart et al., 2013). The eastern parts of South Africa are dominated by thunderstorms during summer months which often result in fatalities, serious injuries, and extensive damage to infrastructure including crops and livestock (Gijben, 2012; NDMC, 2018).

Thunderstorms are primarily classified either as single cell, multicell or supercell – all three types could become very severe and destructive, with supercell thunderstorms notoriously known for producing spectacular tornadoes (Houze, 1993). It should be noted that some multicell thunderstorms can also produce tornadoes (including non-mesocyclonic tornadoes), and that not all supercell thunderstorms produce tornadoes (Markowski and Richardson, 2009). Tornadoes are common around the world, but the most severe are often reported in the United States of America (e.g. Hamill et al., 2005).

South Africa experiences a significant number of severe thunderstorms annually, especially during summer months over the Highveld (Gijben, 2012). A few of these intensify to severe supercell type and often result in strong damaging winds, large hail and tornadoes (Rae, 2014; Glickman, 2000). Tornadoes can occur anywhere in South Africa, but they are frequently

reported in the Highveld regions of Gauteng, north-eastern Free State, western Mpumalanga, and south-central KwaZulu Natal (Goliger and Retief, 2007).

Research has shown that the possible two main important factors for the initiation of tornadoes in thunderstorms are: large amounts of low-level water vapour and the presence of boundary layer vertical wind shear (Markowski and Richardson, 2009). A recent study by Dahl (2017) has shown that the tilting of horizontal vorticity alone is not enough to produce a rotating updraft in supercells. For the tilting of horizontal vorticity, a downdraft is needed possibly at the rear-flank of the storm to pair the updraft in the inflow side of the storm (Davies-Jones et al., 2001). However, this downdraft is not necessary if near-surface vertical vorticity pre-existed before the storm genesis, in which case near surface convergence alone is enough for tornadogenesis (Markowski and Richardson, 2009). Davies-Jones (2015) has summarised tornadogenesis in supercells as occurring in three stages: the formation of mesocyclone in the mid-level, rotation at the ground, and tornado formation or failure.

On 11 December 2017, a supercell thunderstorm associated with large hail, strong damaging winds and a tornado tracked through Vaal Marina, a town in the extreme south of the Gauteng Province of South Africa. The tornadic supercell led to trees being uprooted, destroyed commercial and residential properties, and left about 50 people injured, and over 1100 informal settlement dwellers displaced (SAWS, 2018). The aim of this study is to perform an analysis of this supercell in order to understand the conditions that led to its severity including the formation of a tornado in Vaal Marina. The dynamics and thermodynamics of two configurations of a numerical weather prediction (NWP) model operationally run at the South African Weather Service (SAWS) are also analysed and evaluated to see how they performed in predicting this tornadic supercell.

NWP has advanced over the years (Bauer et al., 2015), with current NWP models operating in the convection-permitting scale where convection schemes are switched off or restricted (Clark et al., 2016). A numerical modelling study by Weisman et al. (1997) found that a 4 km model grid spacing may be sufficient to reproduce mesoconvective circulations and net momentum and heat transport of midlatitude type convective systems. With this understanding the super-parametrization modelling procedure was developed with most of the embedded cloud resolving models running with a grid length of 4 km (Randall et al., 2003; Randall et al., 2016). Other studies, however, argue that models of grid spacing of 1 km or less are the ones adequate to represent dynamics and local processes responsible for triggering convection (see Roberts, 2008, and Bryan et al., 2003). In this study, we consider a model run with a grid length of 4.4km and 1.5km.

**2 Data and methods**

**2.1 Area of interest**

The focus area of this study is in the northern parts of the Highveld region of South Africa, and covers Vaal Marina located in the extreme south of the Gauteng Province, and the storm track in the Free State and Mpumalanga provinces as shown in Fig. 1. The locations of SAWS radars used to track the storm of interest and weather stations where data were analysed for

mesoscale circulations, in this study, are also shown. The storm track as shown in Fig. 1 was identified and reproduced using the Thunderstorm Identification, Tracking, Analysis and Nowcasting (TITAN) and QGIS softwares to mark geographic

90    information system (GPS) locations of the cell for each radar scan (for both radars in Irene and Ermelo, individual and merged for validation), from initiation to dissipation of the storm.

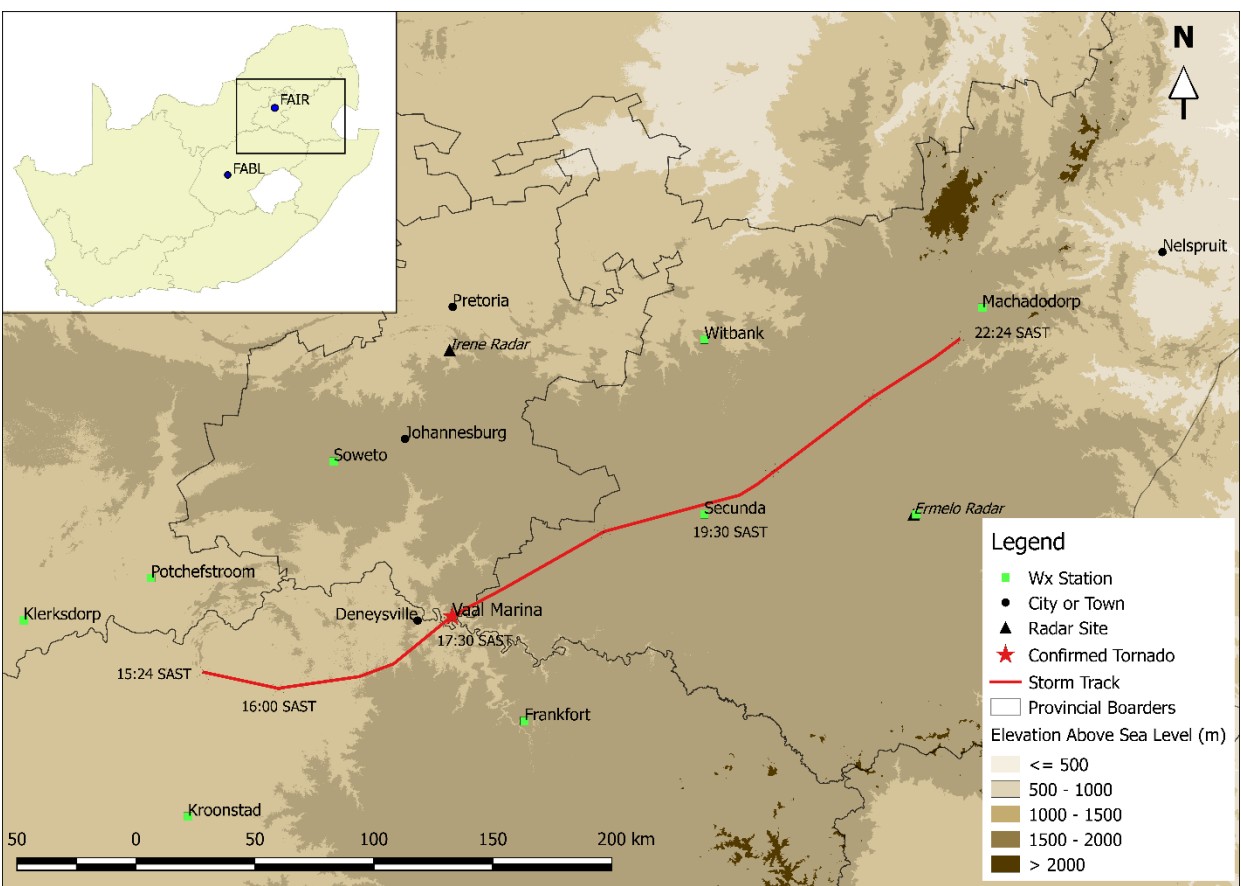

**Figure 1:** The map of South Africa highlighting northern parts of the Highveld (defined as enclosed by the 1500 m above sea level height). The supercell track is indicated with a red line and includes the location of the storm at local times during its 7-hour propagation. Weather stations (Wx Stations) where surface data were analysed for mesoscale circulations are also indicated including the location of the Vaal Marina tornado. The top left embedded image indicate locations where radiosondes analysed were released, in Irene (FAIR) and Bloemfontein (FABL).

**2.2 Datasets**

**2.2.1 The Unified Model**

The primary model data analysed in this study are from the Unified Model (UM) (version 10.4). The UM is a seamless and deterministic numerical modelling system originally developed by the United Kingdom's Meteorological Office (UKMO) for weather prediction and climate studies (Davies et al., 2005). This model is operationally run four times a day on the SAWS CRAY XC-30 high performance computing system. Initial and boundary conditions from the Global Atmosphere (GA, Walters et al., 2017) are used as the driving data for the purpose of producing short-range operational weather forecasts of up to 3 days ahead.

The UM is run in two configurations of the regional and sub-regional domains with 70 vertical levels. The first configuration runs at a horizontal grid length of 4.4 km (this configuration is hereafter referred to as SA4.4) and covers the domain from 0° S to 36° S and 5° E to 54° E. This setup essentially covers the Southern African Development Community (SADC) except the northern parts of Democratic Republic of Congo. The second configuration is centred over South Africa from 20° S to 36° S and 15° E to 34° E and is run at a horizontal grid length of 1.5 km (this configuration is hereafter referred to as SA1.5). The model configurations analysed in this study were both operationally initialised and run at 00:00 UTC on 11 December 2017.

The dynamical core of the model configurations in this study uses a semi-implicit semi-Lagrangian formulation to solve the non-hydrostatic, fully-compressible deep-atmosphere equations of motion (Wood et al., 2014). On the other hand, physical parameterisations of sub-grid processes are split into slow processes (radiation and microphysics) and fast processes (atmospheric boundary layer turbulence, cloud and surface coupling) which are treated and computed differently.

**2.2.2 Radiosonde, satellite, station and radar data**

To perform an analysis of the thunderstorm and its proximal environment during its lifecycle, surface weather station data and radar data are obtained from the SAWS weather station and radar networks. Surface observation data from a total of up to 228 automatic weather stations across South Africa, were regridded and used for synoptic analysis. The regridding is done using the Cressman objective analysis scheme which is described in detail by Cressman (1959). The regridded surface data points are 25 km apart. Only a selected number of stations proximally closer to the storm track in the Free State, Gauteng and Mpumalanga provinces were used for mesoanalysis (see Fig. 1). The surface stations data used contain temperature, dewpoint temperature, pressure, wind speed, wind direction, and rainfall. A surface chart analysis was also performed using data from these stations.

To capture data from middle to lower levels of the storm, the Irene (25.87° S, 28.22° E) and Ermelo (26.53° S, 30.03° E) weather radars are used to track the storm throughout its cycle, from initiation to dissipation (see Fig. 1). We also include the Integrated Multi-satellitE Retrievals for Global Precipitation Measurement (GPM) (IMERG) for precipitation observation (Huffman et al., 2019).

A 1200 UTC Irene weather office's Radiosonde sounding, on the day considered, is used as a proximity sounding of the atmospheric conditions at Vaal Marina. The Irene weather office is located about 110 km north of Vaal Marina and is therefore within the proximity sounding criteria for the location as defined by Craven and Brooks (2004). Radiosonde sounding from the Bram Fischer International Airport (FABL) in Bloemfontein was also used to capture upstreams winds.

### 2.2.3 Reanalysis data and synoptic analysis

A fifth generation European Centre for Medium-Range Weather Forecasts (ECMWF) reanalysis of the atmosphere (ERA5) on single levels and pressure levels, hourly data are utilised to analyse meteorological conditions of the day. The ERA5 dataset is the state-of-the-art atmospheric reanalysis of the global climate with a horizontal grid length of 0.25°×0.25° and 137 vertical

levels up to 1 Pa (Hersbach and Dee, 2016). Parameters analysed from this dataset include 10 m winds, mean sea level pressure, convective and non-convective precipitation, and geopotential heights at 850 -, 500, and 300 hPa levels. A surface chart analysis perfomed by operational forecasters at the SAWS for 12 UTC 11 December 2017 is also used in the synopic analysis and mesoanalysis.

To further analyse instability, ERA5 dataset is used to plot and analyse the environmental lapse rate (ELR), and used to supplement upper

air sounding data. ELR fields are analysed between 700 hPa and 500 hPa, which typically represents the mid-levels of the atmosphere. The mid-level ELRs are calculated by looking at the decrease in temperature with height (Glickman, 2000) as indicated by the following equation:

$$\Gamma = -\frac{dT}{dz}$$

where $T$ is the temperature in °C, $z$ is the altitude in km, and $\Gamma$ is the lapse rate in °C/km.

### 2.2.4 Dynamic and thermodynamic analysis

In this study, both the 4.4 km and 1.5 km horizontal grid spacing of the model configurations are not high enough to reliably use only wind vectors as a determinant of organised mesoscale rotation. As an alternative, vorticity is used as a measure of rotation, and has an added advantage in that it includes most of the flow even in our considered resolutions and is also much easier to use for rotation analysis compared to only using wind vectors (Stevens and Crum, 2003).

Vorticity is a curl of velocity and gives the measure of a fluid's infinitesimal rotation (Holton and Hakim, 2013). Ignoring the effects of friction, the Boussinesq approximation of the vorticity equation can be written as,

$$\frac{\partial \overline{\omega}}{\partial t} = \nabla \times (\overline{V} \times \overline{\omega}) + \nabla \times (B\hat{k}) \tag{1}$$

Where $\overline{\omega} = (\xi, \eta, \zeta)$ and $\overline{V} = (u, v, w)$ respectively represents vorticity and velocity vectors in the horizontal (x-,y-axis) and vertical (z-axis), and B is the buoyancy.

It is not uncommon for thunderstorms to rotate, with supercell thunderstorms spectacularly distinguished by an even stronger and persistent vertically rotating updraft (Doswell and Burgess, 1998; Houze, 1993). Consequently, the current study utilises the vertical component of relative vorticity $\zeta$ as a diagnostic tool of rotation in the vertical axis. It can be derived from Eq. 1 to get,

$$\frac{\partial \zeta}{\partial t} = \hat{k} \cdot \nabla \times (\overline{V} \times \overline{\omega}) + \hat{k} \cdot \nabla \times (B\hat{k}) \tag{2}$$

It should be noted from Eq. 2 that $\hat{k} \cdot \nabla \times (B\hat{k}) = 0$. This implies that buoyancy is not directly responsible for the generation of vertical vorticity and therefore vertical rotation in thunderstorms. Further simplifying Eq. 2 gives,

$$\frac{\partial \zeta}{\partial t} = -\overline{V} \cdot \nabla \zeta + \overline{\omega}_H \cdot \nabla_H w + \zeta \frac{\partial w}{\partial z} \tag{3}$$

where the terms on the right-hand side of the equation respectively represents the advection, tilting and stretching terms.

In this study our dynamical definition of a mesocyclone follows that of Glickman (2000), which defines it as a cyclonically rotating vortex in a thunderstorm associated with vorticity on the order of $10^{-2} s^{-1}$ or greater. The horizontal scale of this vortex is normally somewhere between 2 to 10 km in a thunderstorm.

To analyse the significance of low-level moisture in the event considered, the convergence of moisture flux is integrated from the surface up to 600 hPa level. Moisture flux convergence (MFC) is a useful diagnostic tool for convection initiation as it combines the effects of moisture advection and convergence and can be computed at any atmospheric pressure level (Banacos and Schultz, 2005). The reason we integrate from the surface to 600 hPa, is because we want to capture the significance of the convergence of low-level moisture fluxes. Ndarana et al. (2020) argues that the flow associated with ridging South Atlantic anticyclones, a low-level synoptic driver in our study, changes completely beyond 600 hPa to become sinusoidal and westerly.

To derive an equation for MFC, the conservation of water vapour is used and is further expanded and written in flux form using the continuity equation to get:

$$\frac{\partial q}{\partial t} + \nabla \cdot (q\overline{V}_h) + \frac{\partial}{\partial p}(q\omega) = S = E - P \tag{4}$$

where q is the specific humidity and the second and third terms respectively represent horizontal and vertical moisture flux divergence. S represents the sources and sinks of the air parcel's water vapour evaporation rate (E) and precipitation rate (P), assuming that all condensed water precipitates immediately. MFC is computed by taking the negative of the horizontal moisture flux divergence in Eq. 4, which gives,

$$MFC = -\nabla \cdot (q\overline{V}_h) \tag{5}$$

MFC is then integrated from the surface ($p_s$) to 600 hPa and becomes,

$$MFC^* = -\frac{1}{g}\int_{600}^{p_s} \nabla \cdot (q\bar{V}_h)dp \qquad (6)$$

## 3 Results and discussion

Media reports indicated that on the late afternoon of 11 December 2017, a severe thunderstorm which resulted in extensive
damage, tracked through the extreme south of the Gauteng Province of South Africa (Mitchley, 2017). Some eyewitnesses
reported seeing a tornado close to Deneysville near the Vaal Dam, which later made its way to Vaal Marina and Mamello
informal settlement (all these locations are hereafter collectively referred to as VAM) (Mashaba and ANA reporter, 2017).
Witnesses also reported strong, damaging winds and large hail (SABC, 2017). Visual impacts of the observed severe storm
were also captured by some reports, and indicate uprooted trees, damaged houses and shacks, including damaged infrastructure
such as overhead power lines (Storm Report SA, 2017). From these vast reports, it is clear that this was a high impact weather
system, and therefore important to analyse.

### 3.1 Synoptic setting and meteorological conditions of 11 December 2017

On 11 December 2017, analysis indicates the presence of a ridging anticyclone which extended north of the south-eastern
subcontinent and in the process of breaking-off from the parent Atlantic anticyclone to merge with the south Indian Ocean
anticyclone located east of South Africa (Indicated by letter H over South-West Indian Ocean, SWIO; Fig. 2(d)). The Atlantic
anticyclone began ridging behind a cold front which moved over south of the continent between 08 and 09 December and was
located south-east of the continent on 10 December. The ridging anticyclone, which was situated in the south-eastern
subcontinent on 10 December, led to an influx of warm and unstable low-level moisture from SWIO through the Mozambique
Channel into the eastern and central provinces of South Africa: Mpumalanga, Limpopo, Gauteng, North West and Free State.
The northward-extended, and now breaking, ridging anticyclone on 11 December was coupled with a deepened surface trough
in the central interior of South Africa (Indicated by letter L over South Africa; Fig. 2(d), cf. Fig. 2(c)), resulting in moisture
level increase in the eastern half of the country due to warm tropical inflow of unstable airmass.

## Synoptic Analysis for 11 December 2017 12-UTC

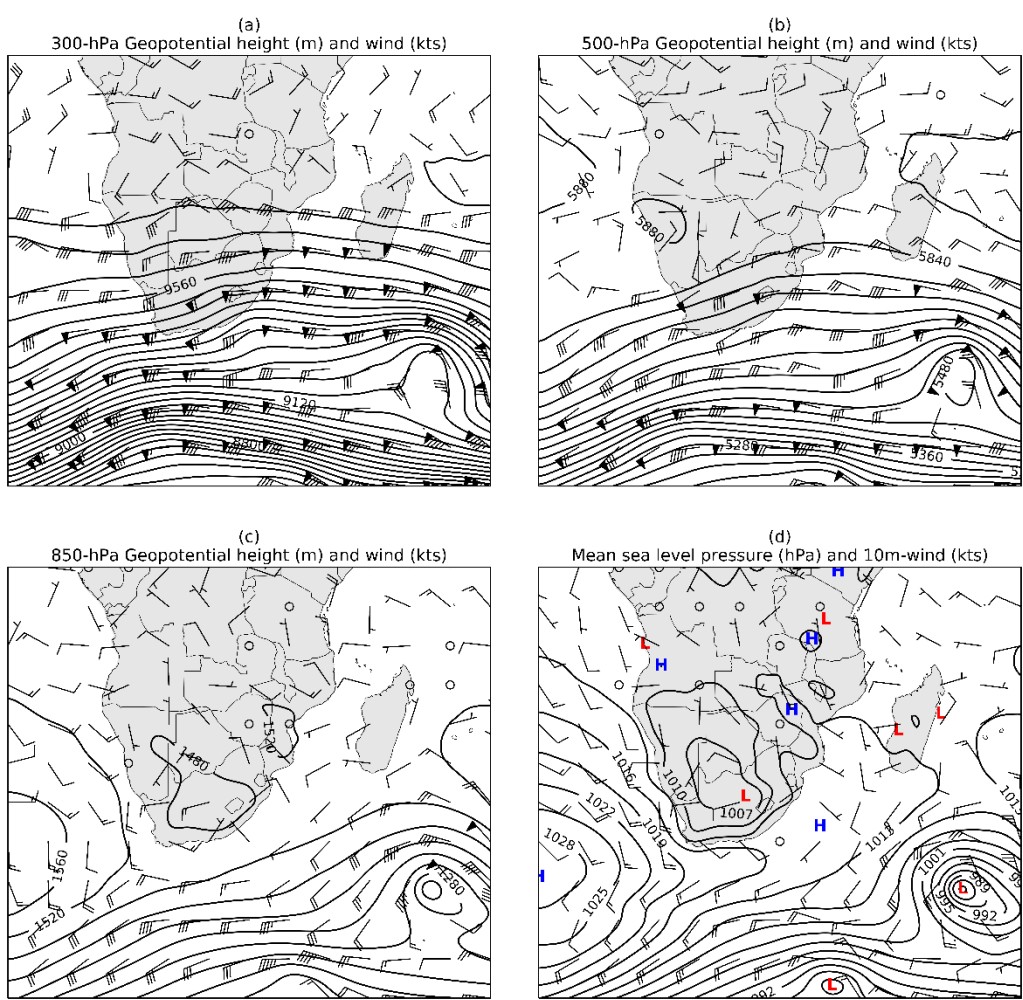

**Figure 2**: Synoptic chart analysis indicating mean sea-level pressure (in hPa) and 850-, 500- and 300 hPa Geopotential heights for 11 December 2017 at 1200 UTC over southern Africa and the surrounding oceans. Descriptions for each frame are contained in the body text.

The analysis also indicates the presence of a mid-tropospheric trough over the south-east of the continent and a ridge extending over Namibia and the western parts of South Africa (Fig. 2(b)). This mid-level circulation pattern resulted in mid-atmospheric south-westerly winds over the interior of South Africa, which enhanced convective instability over the central-east of the country as the dry mid-level air advects over the low-level warm and moist-air in the east. This instability can

clearly be seen from environmental lapse rates which are discussed in much detail in section 3.2 (Mesoanalysis). A weak upper-air trough was also present over South Africa (Fig. 2(a)).

These synoptic settings and conditions (as indicated in Fig. 2) resulted in the western-half of South Africa being hot and dry, while the eastern parts were cool and moist with low-level clouds present (Fig. 3, 4 and 5). Figure 3 indicates that

conditions that prevailed at 12 UTC, also prevailed throughout the afternoon (12 UTC to 16 UTC; 2 pm to 6 pm local time;
Fig. 4 and 5) with little flactuations.

The 12 UTC surface dew-point temperature analysis for 11 December 2017 conducted by operational forecasters at the SAWS, indicated the presence of a dryline over the central Free State Province, which extended through the south-western areas of the North West Province and south-western parts of Botswana (Fig. 3). Figure 4 indicates that at 12 UTC, the reanalysis data compares well with that of the surface chart analysis. It also clearly indicates the dryline over the central Free State
Provice, which also extends through the south-western areas of the North West Province, and south-western parts of Botswana. The dryline separated the hot dry air in the west from the warm moist air in the east, and was moving eastwards after sunrise as differential heating allowed vertical mixing to entrain dry air from the west into the moist boundary layer to the east of the dryline. This dryline was responsible for initiating several severe convective storms in the Free State Province, from which one intensified into a supercell thunderstorm that lasted for 7 hours and resulted in the observed VAM tornado.

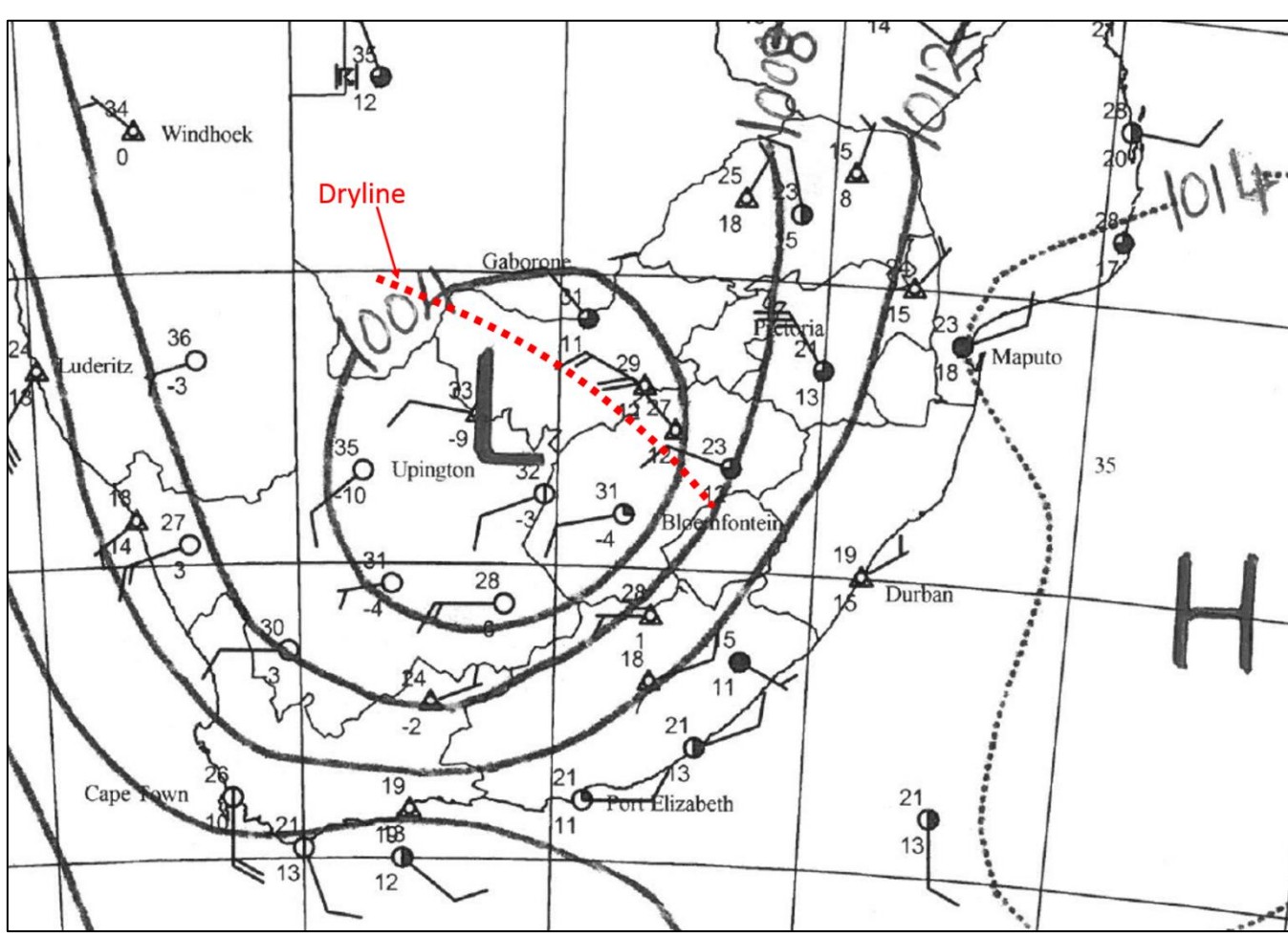

**Figure 3**: Surface chart analysis perfomed at the South African Weather Service for 12 UTC 11 December 2017. The dry line is stretching from the Free State Province to the North West Province.

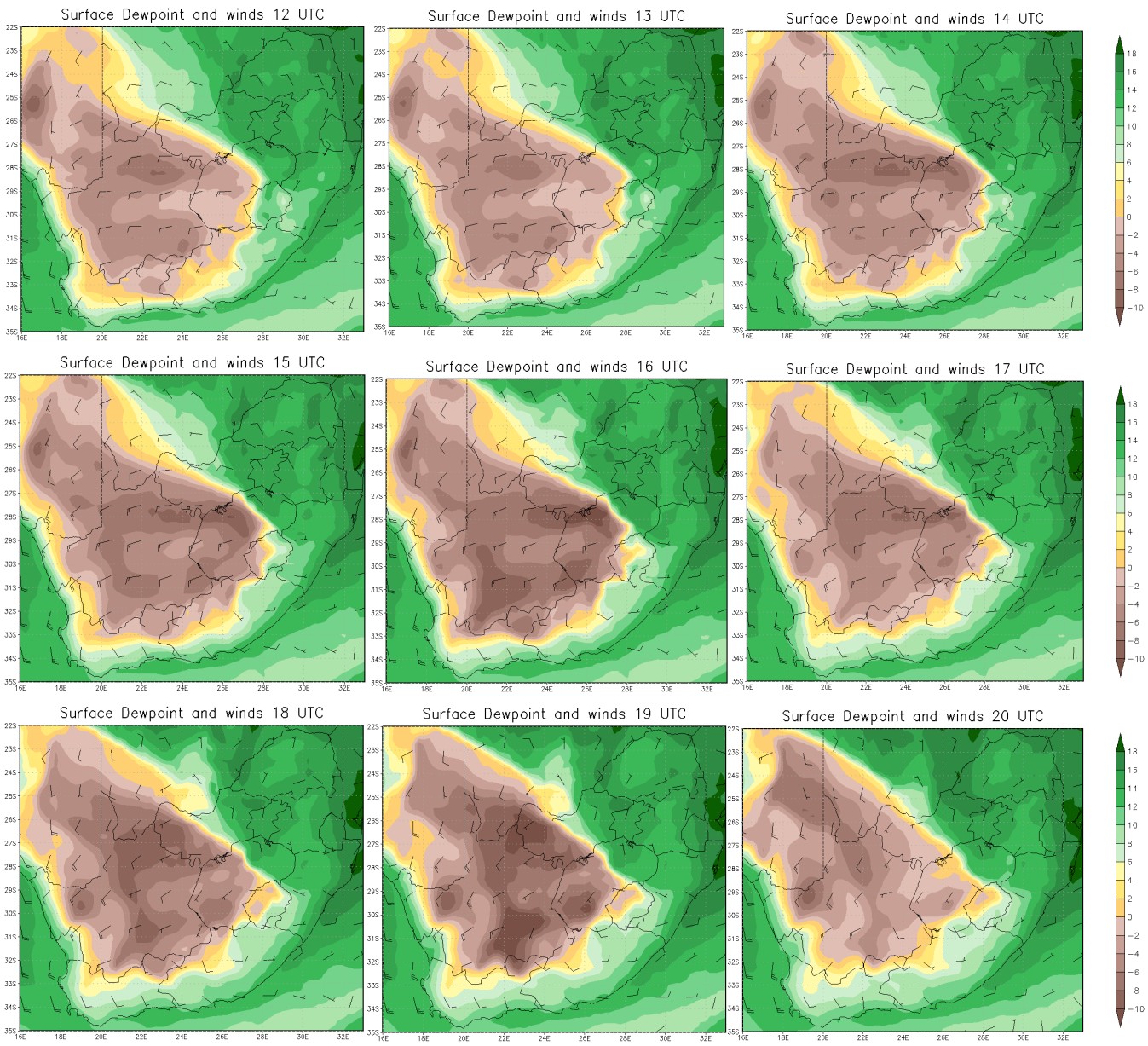

**Figure 4**: Surface dewpoint and winds analysis between 12 UTC and 20 UTC. The dewpoint is in °C. The dryline can clearly be seen in the Free State Province and compares well with that identified by the surface chart analysis in Fig. 3.

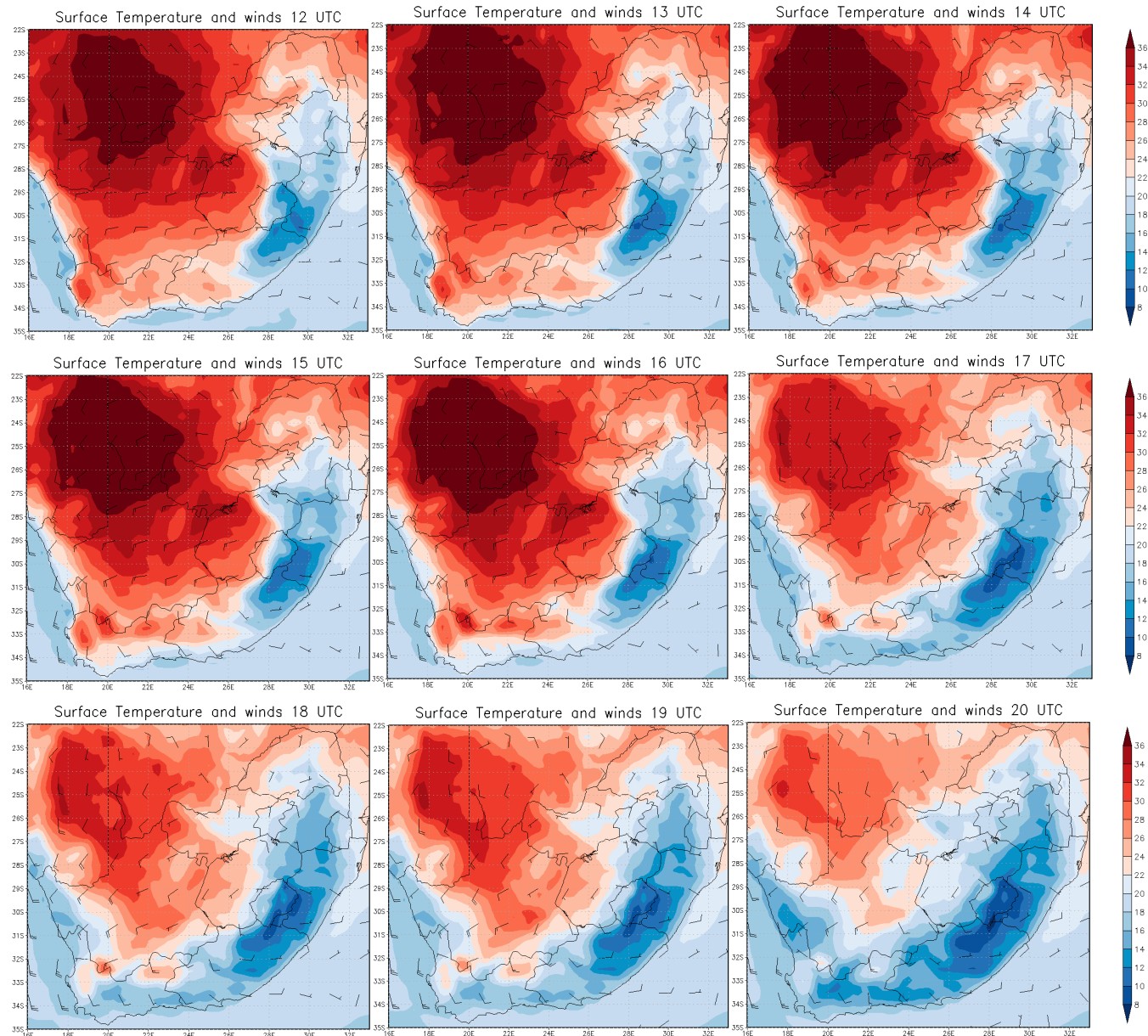

**Figure 5**: Surface temperature and winds analysis between 12 UTC and 20 UTC. The temperature is in °C. The analysis compares well with that of the surface chart analysis in Fig. 3.

**3.2 Mesoanalysis**

Surface station data along near thunderstorm track were used for mesoanalysis. Data analysis from radar and four weather
stations close to the location of the storm initiation (Potchefstroom, Soweto, Klerksdorp and Kroonstad) indicates that the
storm of interest initiated after 13 UTC over a dryline. Figure 6 and 7 indicate that along the dryline, there was a convergence
of moist northerly winds and relatively dry westerly winds. Throughout the analysis period (from 10 to 22 UTC) Potchefstroom
and Soweto recorded an advection of moist air from northerly winds, which respectively averaged 25 and 17 kts between 13
UTC and 14 UTC (Fig. 6 and Fig. 7(a) and 7(b)). During the same analysis period, stations at Klerksdorp and Kroonstad
reported an advection of warm moist air from north-westerly winds which eventually became hot and dry and changed to
westerly after a passage of the dryline (Fig. 6 and Fig. 7(c) and 7(d)). The average hot and relatively dry winds in those stations
between 13 and 14 UTC was 33 kts, with Kroonstad respectively recording 43 and 33 kts.

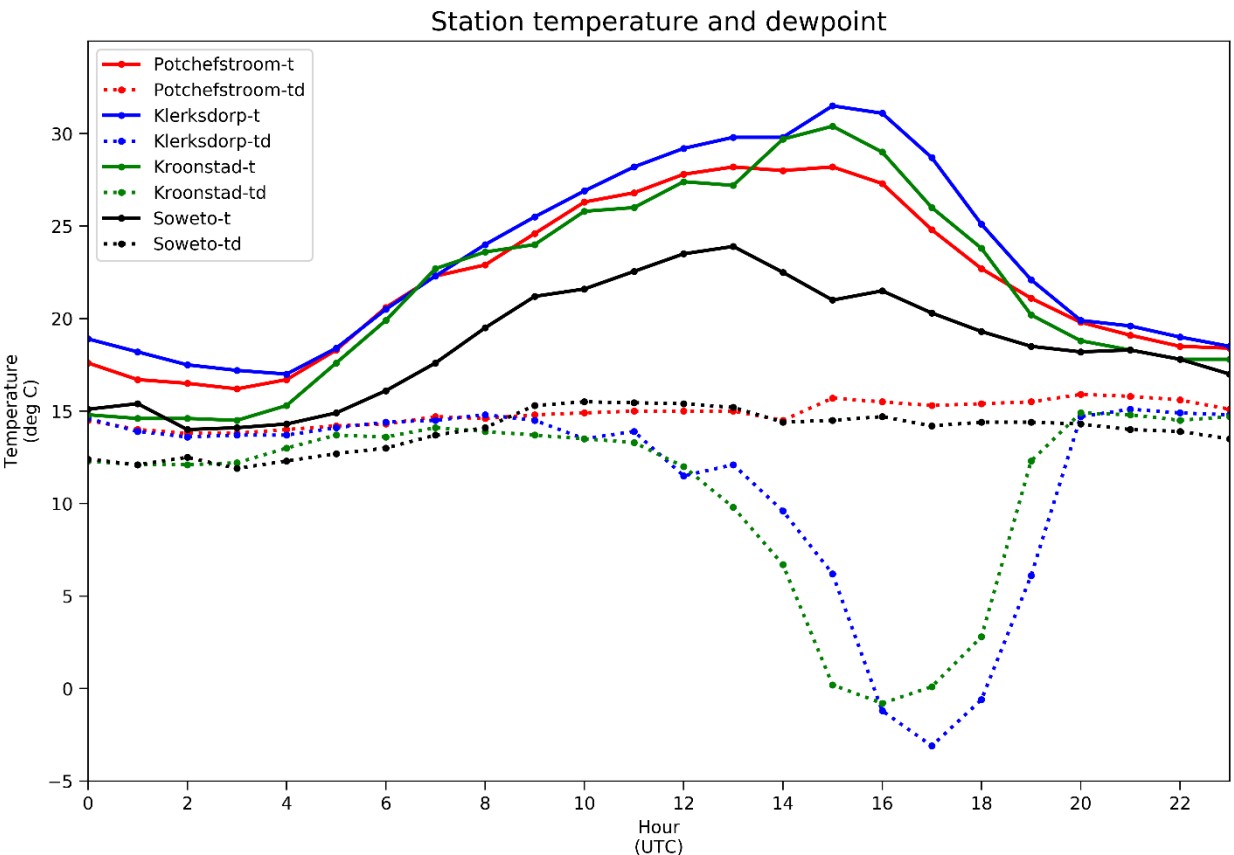

**Figure 6**: The graph is indicating changes in temperature and dewpoint temperature as reported by four weather stations located proximally
closer to the location of the storm initiation between 00 UTC and 23 UTC on 11 December 2017.

After initiation, the storm initially propagated eastwards, then suddenly changed direction to north-east towards the Vaal Dam as it matured into a supercell thunderstorm from a continual merger of several cells (see Fig. 1). As the supercell approached the Vaal Dam, it intensified further and was now associated with large hail and strong damaging winds, with the first sighting of a tornado near Deneysville (town in the north-western edge of the Vaal Dam). The now tornadic supercell then moved across the Vaal Dam and reached Vaal Marina (town in the north-eastern edge of the Vaal Dam), in the extreme south

of the Gauteng Province. In Vaal Marina (and Mamello informal settlement), the supercell was still associated with large hail, strong damaging winds, and a tornado touchdown track of about 1.5 km (1500 m) was confirmed from site visit (and by confirming this in situ data with radar and satellite data).

        It might be significant to note that Frankfort station located a little over 50 km south-south-east of VAM (see Fig. 1) reported cool and moist southerly to south-easterly winds almost throughout the day (Fig. 7(e)) while stations north of VAM

reported cool to warm and moist northerly to north-westerly winds. There is not enough surface data in the vicinity and south of VAM to know whether Frankfort wind speeds and directions generally represent mesoscale circulation at locations south and proximally closer to VAM since it is the only station available around that area. If it does represent mesoscale circulation south of VAM, then it is more likely that there was convergence of cool and moist air close to VAM before tornadogenesis occurred between Deneysville and Vaal Marina. It is important to note this because, studies conducted by Seko et al. (2015)

and Yokota et al. (2016), revealed that low-level water vapour and convergence near the storm are important factors for low-level mesocyclogenesis, which is a process important in supercell tornadogenesis.

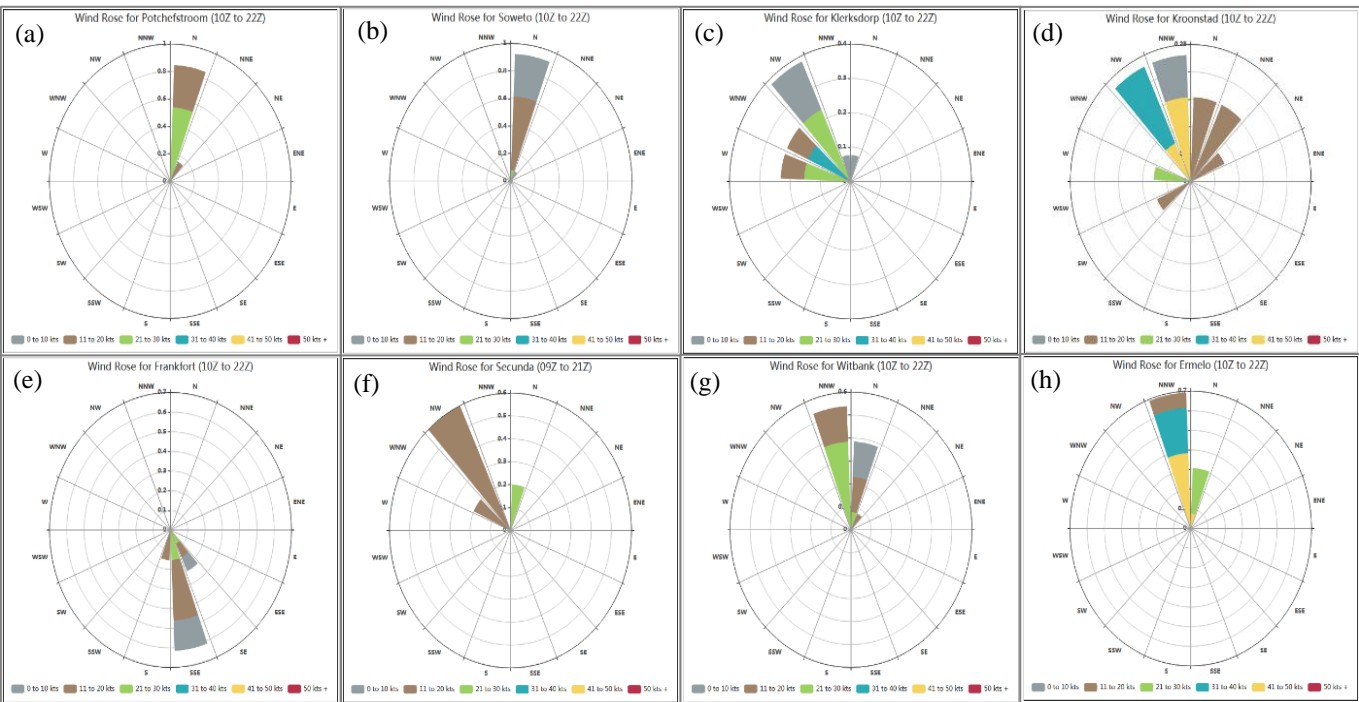

**Figure 7**: Wind rose diagrams of eight weather stations proximally closer to the storm track (as depicted in Fig. 1) between 10:00 and 22:00 UTC on 11 December 2017.

The 12 UTC Irene upper-air sounding (Fig. 8) captured north-north-easterly cool and moist surface winds of 10 kts which back with height (turns counter-clockwise with height, comparative to veering in the Northern Hemisphere) to west-south-westerly at 500 hPa to become 25 kts. The Bram Fischer International Airport (FABL) sounding indicates much greater mid-level winds of 45 kts upstream. The FAIR and FABL upper-air soundings in Fig. 8 confirm winds captured by the synoptic analysis in Fig. 2 close to the soundings vicinity. This indicates significant directional shearing between the surface and the
mid-troposphere, while the speed shearing was respectively weak in FAIR (15 kts at 500 hPa) to moderate (40 kts at 300 hPa). FABL hodograph indicates unidirectional shearing, while that of FAIR is curved and has a common signature for classic supercell development. The FAIR sounding also indicates a 35 kts lower mid-level jet and an upper-level west-north-westerly jet of 65 kts, which also confirm the synoptic analysis in Fig. 2. The lower levels were moist with an average relative humidity of 91% between the surface and 625 hPa, while the mid-levels were dry with relative humidity averaging 26% between 573
270     hPa and 400 hPa.

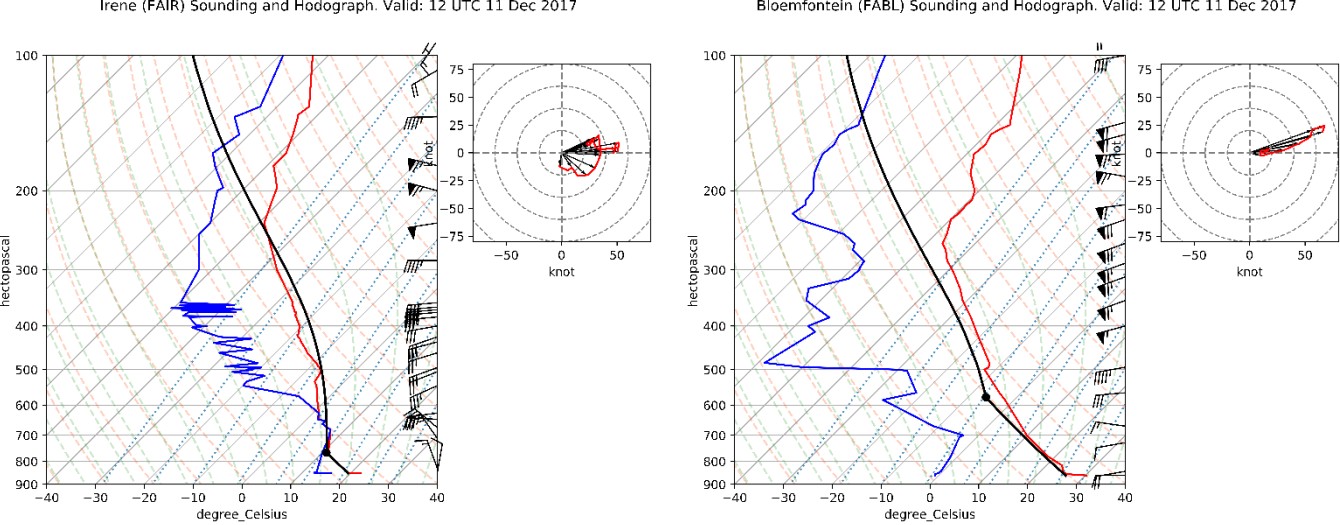

**Figure 8**: Irene (FAIR) upper-air sounding at 1200 UTC on 11 December 2017. This is used to proximate atmospheric conditions over VAM. The Bram Fischer International Airport (FABL) upper-air sounding at 1200 UTC is also included and indicates upstream mid-level winds of about 45 knots. Hodographs are included for each sounding, indicating wind shear between the surface and 10 km above ground level.

ELR fields shown in Fig. 9 indicates that with availabity of moisture and lift, deep convection was favourable in central and eastern South Africa. Between 10 UTC and 14 UTC (12 pm and 2 pm local time), the mid-atmosphere between Irene and VAM had lapse rates values of between 6 and 7 °C/km. These are comparable to lapse rates calculated from the Irene upper-air sounding (FAIR) released at 12 UTC, which was approximately 6.5 °C/km, increasing our confidence that the ELR from the ERA5 reanalysis are reliable. This indicates an atmosphere that was conditionally unstable between Irene and VAM. At the time and location of the storm initiation (15:24 local time (13:24 UTC), cf. Fig. 1), Fig. 9 indicates lapse rates of between 7 and 9 °C/km, implying an unstable atmosphere. As the storm was propagating north-eastward, ELRs along the storm track were increasing. Fig. 9 further indicates that between 10 UTC and 16 UTC (12 pm and 6 pm local time), lapse rates between VAM and Machadodorp ranged between 6 and 7 °C/km. Between 17 UTC and 20 UTC (7 pm and 10 pm) the lapse rates in the same area were now ranging between 7 and 8 °C/km. This analysis indicates an increasingly unstable atmosphere as the storm propagated north-eastward. The availability of low-level moisture, conditionally unstable atmosphere and vertical wind shearing, made it possible for the storm to initiate, intensify to supercell, and then be maintained for 7 hours.

# Mid-level Lapse Rates (°C/km) on 11 December 2017

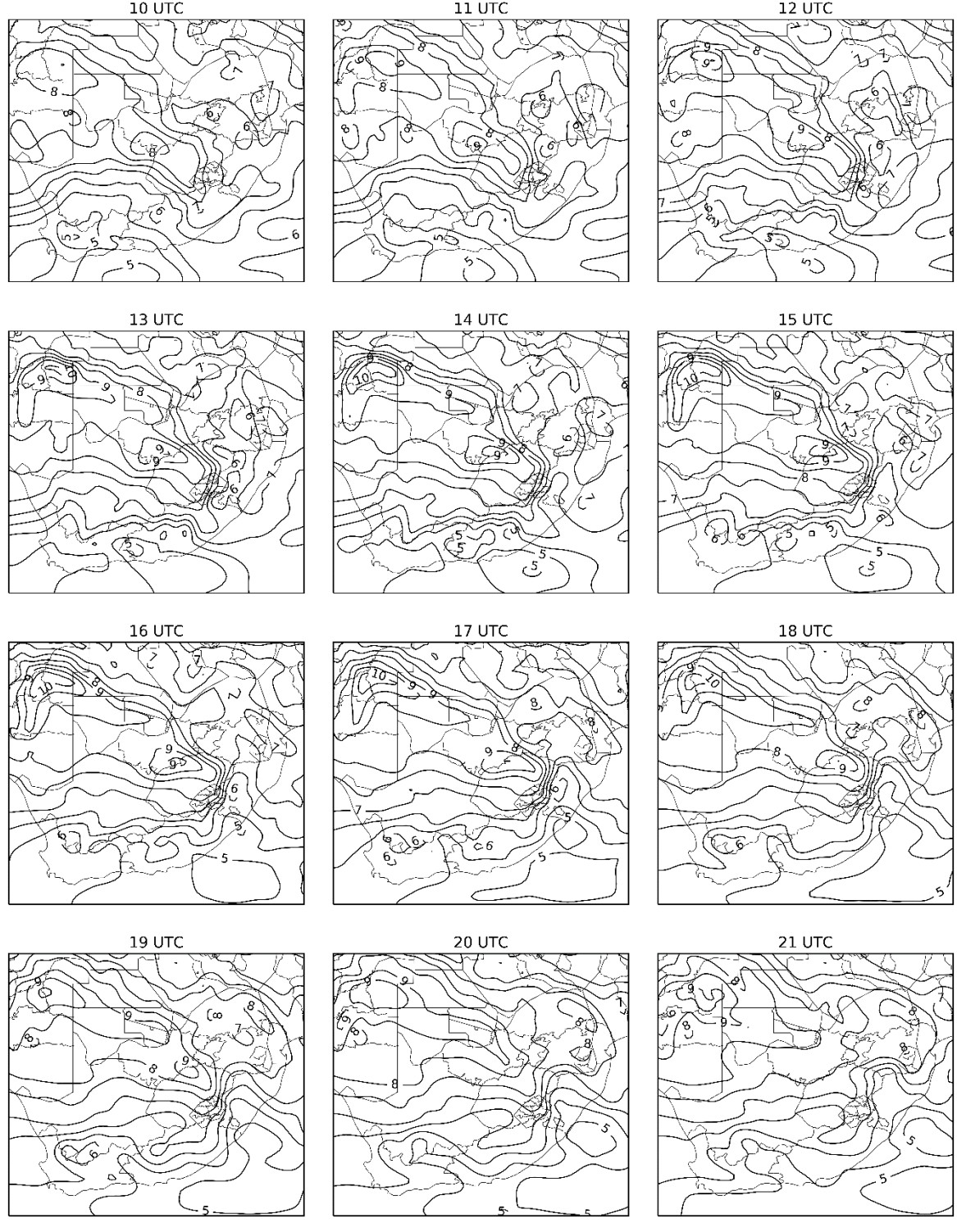

**Figure 9**: Environmental lapse rates (in °C) fields calculated between 700 hPa and 500 hPa (mid-levels) from ERA5 reanalysis. The fields have an interval of 0.5 °C.

Therefore, the availability of low-level moisture, dry mid-levels (which enhanced convective instability), and vertical wind shearing, provided favourable conditions for the possibility of dynamically and thermodynamically induced organised severe thunderstorms (including supercell type) developing in the vicinity. This analysis indicates that three ingredients important for strengthening and maintaining typical supercell thunderstorms were present, and can be summarised for this case as follows: significant surface to mid-level vertical shear, an abundance of low-level warm moisture influx from the tropics

and Mozambique Channel, and the relatively dry mid-levels which enhanced convective instability (specifically builds potential instability) as the dry mid-level air advects over low-level warm and moist-air. Prevailing synoptic environmental conditions further shows that these ingredients also prevailed in the eastern parts of the country, over the Highveld region (see Fig. 1), and therefore along the vicinity of the storm track, which indicates that they likely played a role in ensuring that the supercell storm is strengthened and maintained throughout its lifetime.

A hook echo can clearly be seen from the low-levels of the storm between 15:16 UTC and 15:34 UTC as observed by the first sweep of the Irene radar (at an elevation angle of 0.5°) (Fig. 10). Figure 10(a) indicates a hook echo west of the Vaal Dam at 15:16 UTC which was associated with local strong cyclonically rotating winds as indicated by the radial velocity inbound-outbound maxima in Fig. 10(b) (this could explain the tornado observed near Deneysville). A hook echo was also observed 12 minutes later at 15:28 UTC over Mamello, an informal settlement located in the eastern outskirts of Vaal Marina

(Fig. 10(e)). Figure 10(f) indicates that this hook echo over Mamello was also associated with local strong circulating winds. This analysis indicates that it is most likely that the tornado that was reported to have devastated VAM touched down multiple times between 15:15 and 15:30 UTC (17:15 and 17:30 local time); but could also have been on the ground during this entire period or for a period longer than the mesocyclone that was captured by the radar.

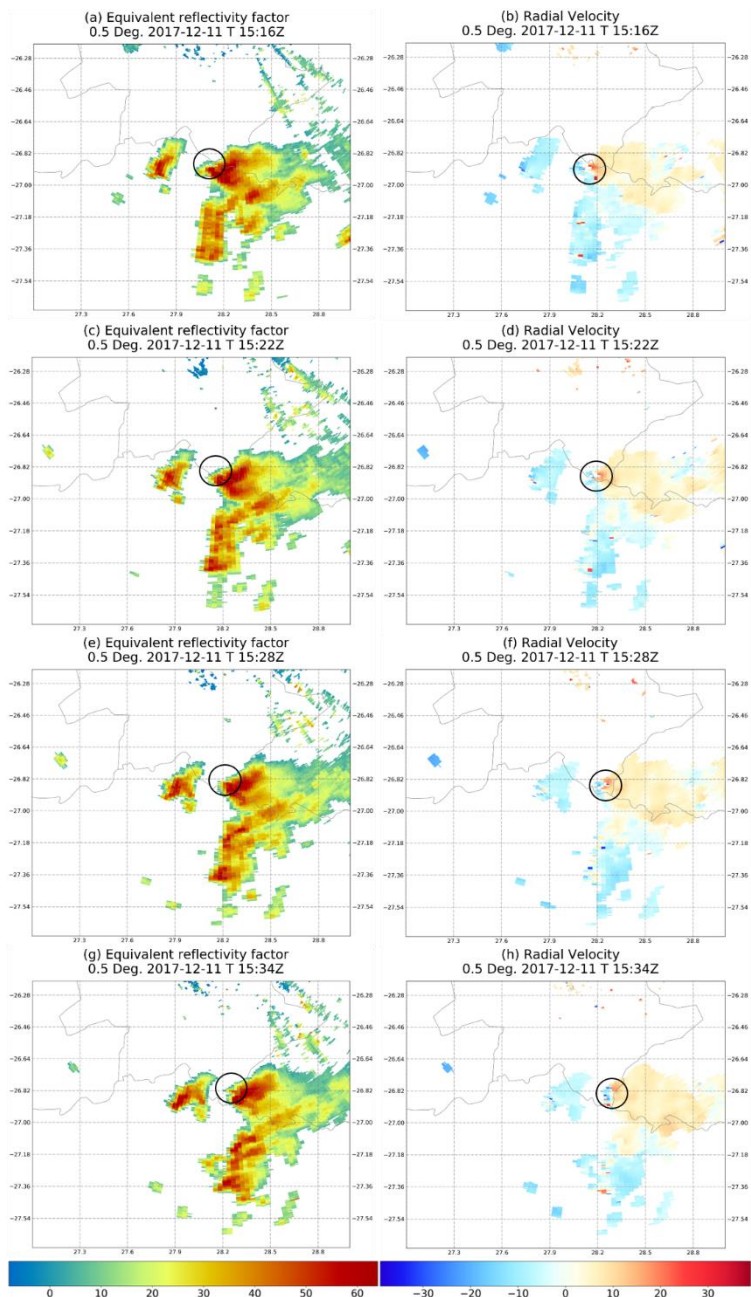

**Figure 10**: Irene radar equivalent reflectivity factor (dBZ) and radial velocity of scatterers (m/s) of the supercell thunderstorm as plotted using Python ARM Radar Toolkit (Py-ART) (Helmus and Collis, 2016). The left-hand column indicates low-level reflectivity as captured by the first radar sweep at an elevation angle of 0.5° for every 6 minutes, between 15:16 UTC and 15:34 UTC. The right-hand column radar scans have similar properties as the left-hand column radar scans but for the radial velocity of scatterers.

After passing through the VAM area, the supercell continued in the north-easterly direction, weakening and strengthening throughout, before dissipating just before reaching Machadodorp in Mpumalanga Province. Throughout near storm track (except for Frankfort), surface winds were variable between northerly and north-westerly (Fig. 7). The supercell dissipated on approach to Machadodorp where winds were variable between north-easterly to easterly between 19 and 20 UTC (21 and 22 local time) and averaged 8 kts.

### 3.3 Unified Model analysis

The twenty-four hour rainfall total is shown in Fig. 11 for both SA1.5 and SA4.4, as well as the ERA5 reanalyses and IMERG which represents observations. Both the ERA5 and IMERG captured rainfall in the area where the storm passed, however there are differences in the two datasets. For example, IMERG indicates a large amount of rainfall in the north-east border of the plot. The ERA5 shows large amount of rainfall only in the south-east of the domain shown. The SA4.4 simulated a large amount of rainfall in different parts of the domain, while the SA1.5 only simulated a large amount of rainfall in the south-east border of the domain. We also investigated the timing of rainfall in the SA4.4, and found that the model predicted a precipitating storm propagating north of VAM between 15 UTC and 16 UTC. The timing of this precipitating storm as predicted by SA4.4 was correct; however, the location was slightly north of the actual observation (cf. Fig. 10). This is a common issue with convection-permitting models which prompted the scientific community to develop new metrics to verify these types of models to avoid penalising them twice for misplacing rainfall (e.g. Davies et al. 2008; Gilleland et al., 2009). On the other hand, SA1.5 failed to predict any precipitation close to VAM between 15 UTC and 16 UTC or anytime during the analysis period (between 12 to 24 UTC) (see Fig. 11).

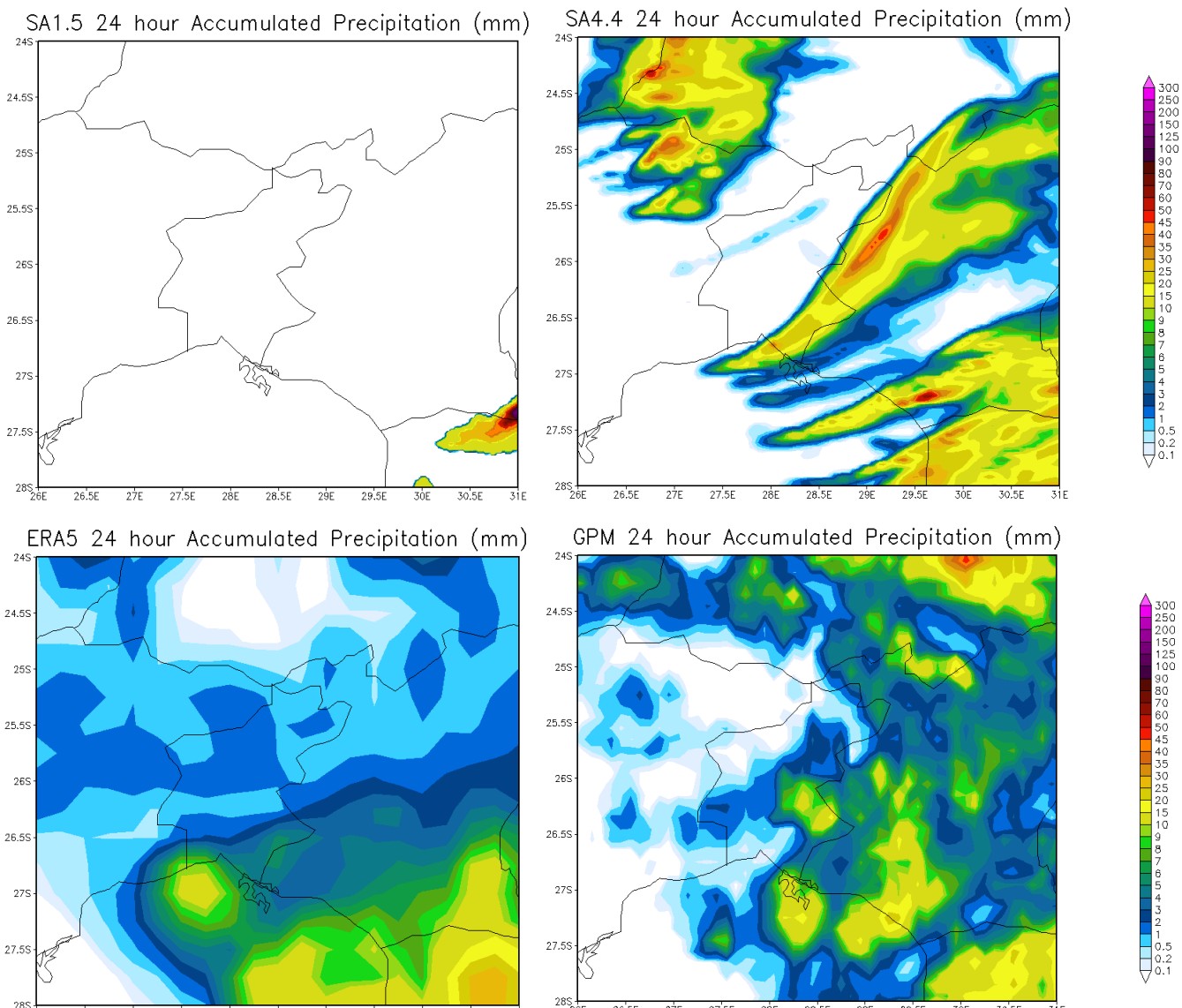

**Figure 11**: Total precipitation for 11 December 2017, in the area of interest (in mm) as predicted by SA1.5 and SA4.4, and also from ERA5 and GPM.

An analysis of the 10 m wind directions in SA4.4 and SA1.5 generally agrees with observed surface northerly to north-westerly wind directions close to VAM, however the wind speeds are underestimated in both models (Fig. 12). Figures 12(a) and 12(b) shows that SA1.5 and SA4.4 predicted almost similar wind speeds at 15 UTC, which are underestimated compared to observations (Fig. 12(d)). The simulated wind speeds by the UM is more similar to the ERA5 (Fig. 12(c)) , and the similarities are larger with the SA4.4. The ERA5 winds are also lower than those observed using SAWS ground stations. A

sample of three stations along the storm track taken proximally closer to the location of initiation, propagation and dissipation stages of the storm (Klerksdorp, Witbank and Machadodorp) indicate that SA4.4 and SA1.5 especially underestimated the wind speeds during the initiation stage of the storm (Fig. 13). The ERA5 winds are stronger than both UM configurations, despite the ERA5's lower resolution compared to these configurations.

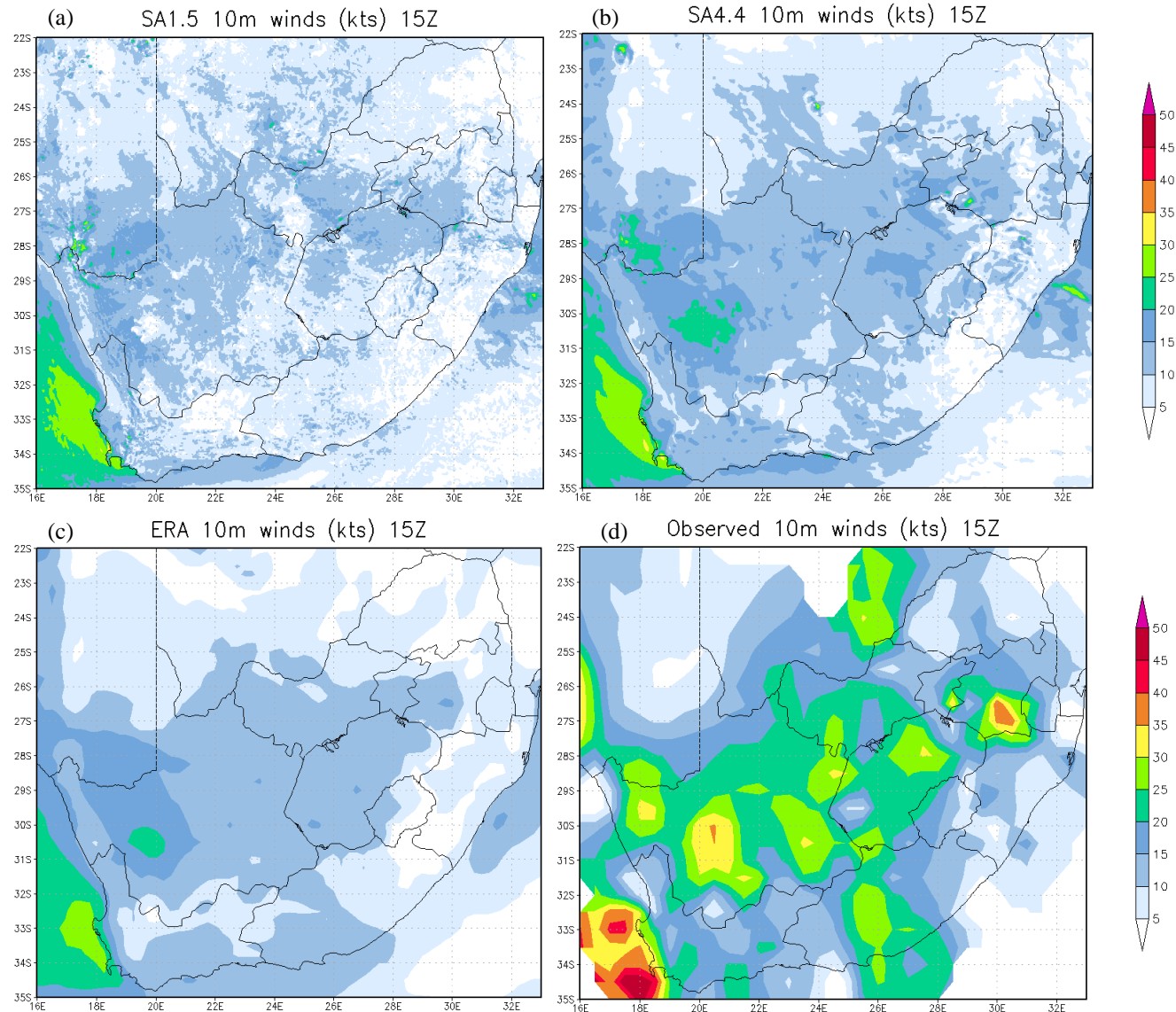

**Figure 12**: A comparison of observed surface wind speed from weather stations across South Africa at 15:00 UTC (d) with 10 m predicted winds by (a) SA1.5 and (b) SA4.4. Figure (c) indicates ERA5 winds. All the wind magnitudes are given in knots at an interval of 5 knots.

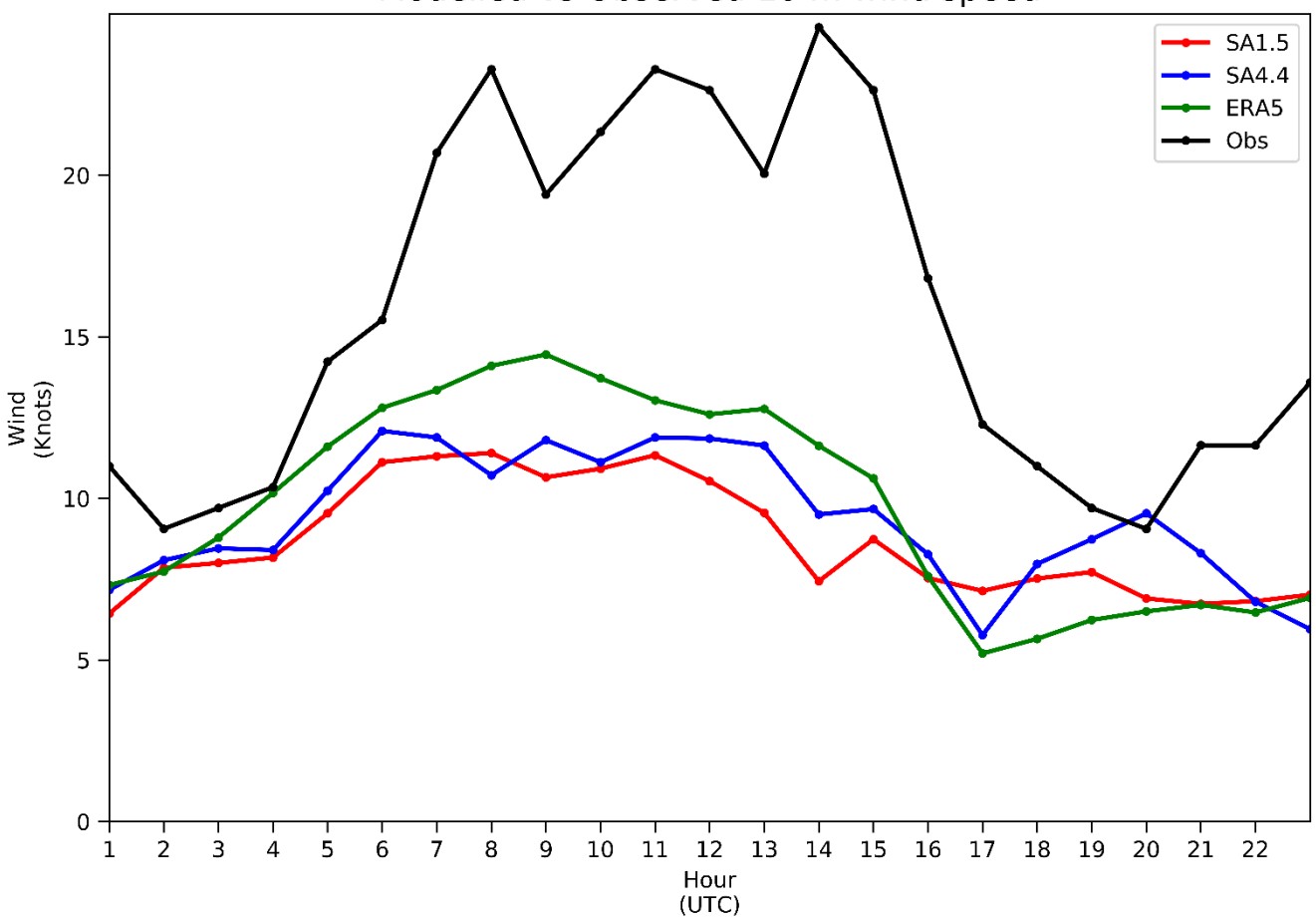

**Figure 13**: Figure indicates a comparison of average wind speeds from three stations along the storm track (Obs) compared to SA1.5, SA4.4, and ERA5 data for those stations between 01:00 UTC and 23:00 UTC. All the wind magnitudes are given in knots at an interval of 5 knots.

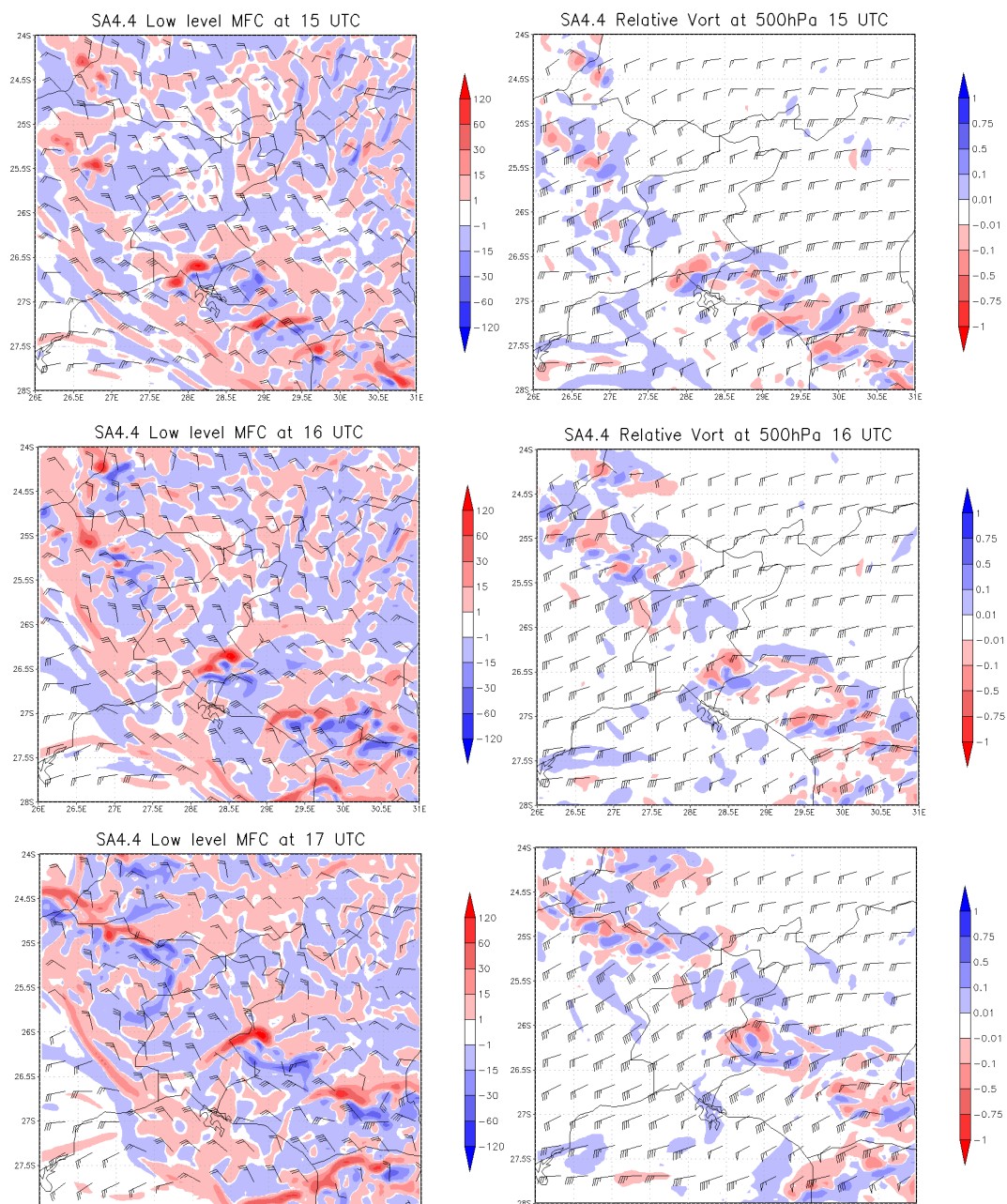

**Figure 14**: Low-level MFC and 500 hPa relative vorticity and wind barbs as depicted by SA4.4. The shades in the relative vorticity figures represent negative (red) and positive (blue) relative vorticity in $10^{-2}\ s^{-1}$. MFC is in $10^{-6}\ g\ kg^{-1}s^{-1}$, with positive values (red shades) representing areas of convergence of moisture and negative values (blue) represents areas of divergence of moisture.

The SA4.4 generally indicates westerly and south-westerly 500 hPa winds at 15 UTC over VAM (Fig. 14). Considering surface winds, the SA4.4 indicates the presence of vertical shear between the surface and the mid-levels, however the speed shearing is possibly underestimated especially during the initiation phase of the storm. The SA4.4 predicted a south-westerly propagation of 500 hPa vertical vorticity north of VAM between 15 UTC and 16 UTC (Fig. 14). The maximum negative vorticity at 500 hPa reached by the VAM storm, as predicted by SA4.4, ranged between $-0.5 \times 10^{-2}\ s^{-1}$ and $-0.75 \times 10^{-2}\ s^{-1}$ from 16 UTC and 17 UTC. This vertical vorticity was one order of magnitude smaller than a dynamical definition of a mesocyclone, which is an indicator that SA4.4 did not capture the mesocyclone of the VAM storm, and therefore underestimated its severity. However, contrary to supercell storms in the midlatitudes, most storms developing in the subtropics (which is where South Africa is located) do so without strong vertical shear and are rather associated with weak or moderate vertical wind shear (Sansom, 1966; Hand and Cappelluti, 2011). It may therefore be argued that relatively lower amounts of vertical wind shear in the subtropics, and in our case study, correlates with relatively lower relative vertical vorticity, and therefore relatively weaker mesocyclone according to classical definitions as stated by Glickman (2000).

A relationship between MFC and vertical vorticity was also determined. It was found that SA4.4 predicted vorticity maxima at 500 hPa that is associated with MFC maxima at low-levels. MFC initiates first at the low-level followed by vorticity genesis at 500 hPa. This implies that the development of MFC in the low-levels could be a precursor of vertical vorticity initiation in the mid-levels. Since there is a positive relationship between low-level MFC and mid-level vorticity, it follows that the stronger the low-level MFC, the stronger the mid-level vorticity would be. This finding is in agreement with a study by Banacos and Schultz (2005)

Further examination of the VAM storm during its 500 hPa vorticity maxima as predicted by SA4.4 (which occurred between 16 UTC and 17 UTC), reveals that low-level convergence occurred in the equatorward side of the storm, and divergence in the polarward side of the storm. Figure 15(a) indicates that low-level convergence was associated with relatively higher values of specific humidity. This implies that there is an inflow of low-level moist air which converges in the equatorward side of the storm and then gets uplifted as updrafts. Figure 15(a) also indicates that low-level divergence is associated with relatively lower values of specific humidity which implies that there is low-level diverging dry-air in the polarward side of the storm. Figure 15(b) shows that the low-level diverging dry-air depicted in Figure 15(a) particularly originates from the mid-levels.

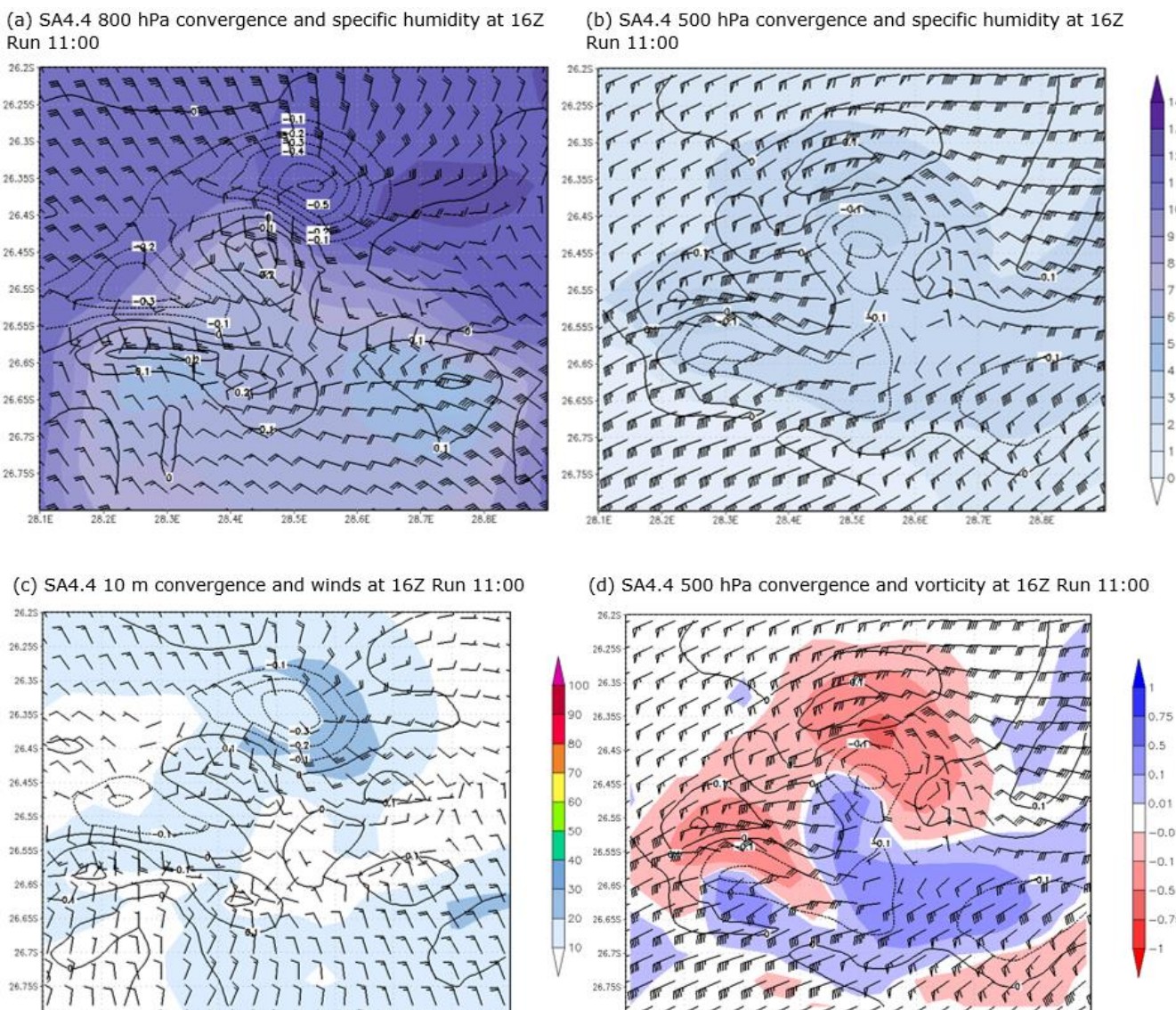

**Figure 15**: 16:00 UTC SA4.4 analysis of the (a) 800 hPa and (b) 500 hPa convergence/divergence (in $10^{-2}s^{-1}$) at contour intervals of $0.1 \times 10^{-2}s^{-1}$ and specific humidity (in $g\ kg^{-1}$) at shaded fields of interval $1 \times g\ kg^{-1}$. (c) Indicates 10 m convergence/divergence contour fields (in $10^{-2}s^{-1}$) at $0.1 \times 10^{-2}s^{-1}$ intervals and wind barbs and wind magnitude shades (in knots) at 10 knots intervals. (d) Indicates 500 hPa convergence/divergence (in $10^{-2}s^{-1}$) at contour intervals of $0.1 \times 10^{-2}s^{-1}$ and vorticity (in $10^{-2}s^{-1}$) shaded fields.

The analysis also indicates a coupling of 800 hPa convergence (divergence) fields with 500 hPa (divergence) convergence fields, which is an indicator that the low-level converging moist-air in the equatorward side of the storm gets uplifted with updrafts to the dry mid-levels then later gets advected to the low-levels with the downdrafts in the polarward side of the storm (Fig. 15(a) and 15(b)).

Figure 15(c) depicts 10 m winds and convergence (divergence) analysis. It should be noted that this analysis indicates that near-surface convergence (divergence) field patterns are similar to those at 800 hPa shown in Fig. 15(a), of which are together coupled with 500 hPa divergence (convergence) fields shown in Fig. 15(b). Figure 15(c) indicates that there are two areas of updrafts origin near the surface (ahead of the storm and at the rear-flank of the storm) which are both located in the equatorward side of the storm, as also confirmed by the mid-level convergence and vorticity analysis (Fig. 15(d)). Figure 15(c) also indicates areas of forward-flank and rear-flank downdrafts which are in the polarward side of the storm as also confirmed by the mid-level convergence and vorticity analysis (Fig. 15(d)).

Figure 15 can be summarised as follows. There is a convergence of moist and relatively stronger surface and low-level winds ahead of the storm which are uplifted as updrafts to the mid-levels. Some of these updrafts diverge as they reach the mid-levels and are later advected to the lower-levels and surface as relatively dry downdrafts, which then also diverge ahead of the downdraft area (toward the north-east as part of a front-flank gust front) and toward the rear-flank of the storm (toward the west as part of a rear-flank gust front) on reaching the surface. The north-eastward diverging winds then contribute to uplifting the inflowing moist air ahead of the storm. On the other hand, the westward diverging winds converges with north-westerly inflowing moist-air in the rear-flank of the storm, initiating a second area of updrafts in the rear-flank of the storm. This rear-flank updrafts also diverge in the mid-levels and contribute to the forward-flank downdrafts. The diverging mid-level winds from the rear-flank updrafts also converge with mid-level dry-air which is being entrained from behind the storm and are later advected to the surface as the rear-flank downdrafts.. Figure 15 further indicates that surface and low-level convergence is associated with the storm's cyclonic updrafts, while divergence is associated with anticyclonic downdrafts.

It is interesting to note that this analysis indicates that SA4.4 predicted thunderstorm dynamics consisted with those of a classic supercell. Therefore, from a dynamical perspective, SA4.4 was able to predict the VAM supercell, even though its strength was underestimated, and location incorrect possibly due to a lack of data assimilation.

### 3.4 Summary and conclusion

In this study an analysis of the tornadic supercell that tracked through the northern parts of the Highveld in South Africa on 11 December 2017 was performed. It was found that this supercell initialised in the Free State Province as part of a cluster of multicellular thunderstorms over a dryline, and propagated in a north-easterly direction while strengthening and weakening throughout its lifetime, until it dissipated 7 hours later on approach to Machadodorp in the Mpumalanga Province. It was also found that three ingredients were likely important in strengthening and maintaining this supercell: significant surface to mid-level vertical shear, an abundance of low-level warm moisture influx from the tropics and Mozambique Channel, and the relatively dry mid-levels which enhanced convective instability.

On approach to the extreme south of the Gauteng Province, the already severe supercell encountered even more low-level moisture in the area which resulted in a tornado being initiated south of Deneysville town at approximately 17:15 local time. The supercell tracked through the Vaal Dam, and continued through the town of Vaal Marina and the Mamello informal settlement at around 17:30 local time.

NWP models operationally run at the SAWS were also analysed and evaluated to see how they performed in predicting this tornadic supercell. It was found that the 4.4 km grid spacing model (SA4.4) performed better than the 1.5 km grid spacing model (SA1.5) in predicting this supercell. This is in spite of the horizontal resolution of SA1.5 being higher than that of SA4.4, and therefore, expected to be able to represent small-scale atmospheric processes better. The poor performance of SA1.5 might have emanated from a significant underestimation in low-level warm moisture advection and convergence (which act as dynamic lifting mechanisms). Alternaticely, it could be a case of missed convection initiation. Keat et al (2019) have found that most rainfall, as simulated by SA1.5, is produced by large storms of at least 50 km in diameter. In our study, radar indicates that the storm being analysed had a diameter of between 20 km and 30 km, which may provide an explanation of why SA1.5 does not capture any storm, and therefore may be a case of missed convection initiation.

SA4.4 was able to capture the supercell being analysed. However, the severity of this supercell was underestimated, possibly due to an underestimation in the mid-level vorticity which was found to be one order of magnitude smaller than that of a typical mesocyclone. This underestimation in mid-level vorticity might be a result of poor model resolution of surface to mid-level vertical wind shear and low-level horizontal mass and moisture flux convergence, due to the model grid-spacing. The poor model resolution is, therefore, the first possible explanation of why SA4.4 might have underestimated the mesocyclone. The second possible explanation is found in previous studies conducted by Sanson (1966) and Hand and Cappelluti (2011). These studies note that vertical wind shear in subtropical regions is usually weaker compared to mid-latitude regions. This might therefore result in weaker vorticity and mesocyclones. Further studies will be required to ascertain this and test the traditional dynamic mesocyclone definition by Glickman (2000) for subtropical regions. SA4.4 also predicted a correct timing of the supercell, however the location was incorrect which is a common issue found in convective permitting models. This result is consistent with those from studies by Stein et al. (2019), which found that while the correct cover of rainfall may be predicted by SA4.4, it may be predicted in the wrong location.

From these results, it is recommended that research into the possibility and benefits of implementing dynamic and thermodynamic objective analysis schemes, which are derived from both NWP model data output and near real-time surface and upper-air observations, be conducted for a very short-range objective prediction of severe thunderstorm (including supercells) initiation over the central and eastern parts of South Africa. It is also recommended that the classical dynamic definition of mesocyclones be tested for subtropical regions, where vertical wind shears are typically weaker compared to the mid-latitudes.

Future investigations will involve experimental research over the Highveld region of South Africa to understand mesoscale and local dynamics processes responsible for tornadogenesis in some severe storms. Such a study, to the best of our knowledge, has never been conducted. This will help improve representation of regional and local processes in mesoscale NWP models which will in turn improve the skill of severe storm prediction over South Africa.

*Data availability.* The UM is licensed and legal possibilities of accessing its operational data should be cleared with the SAWS. Radiosonde data can be obtained from the Integrated Global Radiosonde Archive Version 2 (IGRA 2) via https://doi.org/10.7289/V5X63K0Q. Radar and weather stations data can be obtained from the SAWS. ERA5 data can be obtained from Copernicus Climate Change Service Climate Data Store via https://climate.copernicus.eu/climate-reanalysis.

*Author contributions.* LEL and MMB conceived and designed the study. LEL, GR, MMB and PM performed the analysis while continuously discussing results with all authors who in turn contributed through their feedback. GR, MG, LEL and NM curated the data. LEL wrote the initial draft of the manuscript with all authors reviewing and contributing in various proportion to sections. All authors edited the final draft and provided significant comments and suggestions for further clarity and improvements.

*Competing interests.* The authors declare that they have no conflict of interest.

*Acknowledgements.* The authors thank the SAWS for providing model and observational data for this study. This work was partially supported through the Climate Research for Development (CR4D) Fellowship implemented by the African Academy of Sciences (AAS) in partnership with the United Kingdom's Department for International Development (DfID) Weather and Climate Information Services for Africa (WISER) programme and the African Climate Policy Centre (ACPC) of the United Nations Economic Commission for Africa (UNECA). The authors also acknowledge the SAWS for supporting this study.

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
