# Peer review of "A dynamic and thermodynamic analysis of the 11 December 2017 tornadic supercell in the Highveld of South Africa"

_Weather and Climate Dynamics, 2020_

## Referee Comment (RC2)

**Review of WCD-2020-34**

**Authors:** Lesetja E. Lekoloane et al.
**Title:** "A dynamic and thermodynamic analysis of the 11 December 2017 tornadic supercell in the Highveld of South Africa"
**Recommendation:** Potentially acceptable pending major revisions

**Overview**
Lekoloane et al. provide an interesting case study and a pair of model simulations associated with the 11 December 2017 tornadic supercell in the Highveld of South Africa. While this case would be of interest to the community, I have some serious reservations about the manuscript in its current form. In particular, there appear to be some fundamental misunderstandings—or at least omissions—related to the current knowledge base of supercell tornadogenesis. Additionally, I am concerned that the analyses of high-resolution simulations are too limited in scope in their current state and also reveal some misunderstandings. These issues could be alleviated with considerable revision and reanalysis, but I do not think the manuscript is suitable for publication at this time.

**Major Comments**
1. Lines 57-59: Your discussion about near-surface vertical vorticity is short and potentially contains some misunderstandings. Tilting of environmental horizontal vorticity is the primary contributor to updraft rotation within a supercell, but it is not typically thought to contribute significantly to near-surface vertical vorticity. The other process that you mention could achieve near-surface vertical vorticity ("advection of vorticity from aloft") is vague, and I cannot be certain that what you mean here is actually consistent with current thinking. Please revise this section for clarity, and if necessary, review recent literature on the topic to ensure your introduction is consistent.

2a. Your method of evaluating model simulations is potentially flawed. SA1.5 very clearly lacks a supercell, as there was no precipitation present in the specific time frame. Thus, the use of midlevel vorticity was a strange choice to determine the presence of a supercell (or lack thereof). Additionally, you look at one hour here; could the higher-resolution simulation simply be delayed in time? What happened beyond 16z? This analysis seems to be somewhat cursory, and perhaps too superficial for a peer-reviewed manuscript.

2b. Your comment on lines 342-343, "…as a result of the mid-level vorticity being greatly underestimated, which led to the storm not being initialized" shows a fundamental misunderstanding of the source of modeled midlevel vorticity, or at least inconsistency with your analysis of SA4.4. The midlevel vorticity maxima/minima couplets in these simulations appear to be a *consequence* of the development of convection, not a cause. Please re-evaluate this statement.

2c. Likewise, I have some concerns about your analysis and interpretations of SA4.4. As you noted, this simulation likely did not resolve the mesocyclone adequately, which is expected given the fairly coarse resolution (for a CAM). There are also some claims not entirely supported by the data provided, discussed below.

"…it follows that the stronger the low-level MFC, the stronger the mid-level vorticity would be." While this is likely just consequence of convergence below an updraft and the corresponding

updraft, I'm not sure that the data support this claim in a more general sense. Do you have more evidence?

Lines 317-318: Given that SA4.4 doesn't adequately resolve the mesocyclone, I don't think you can expect it to accurately represent downdraft processes. RFDs tend to be considered dynamic features associated with low-level rotation within the storms. If the rotation isn't being resolved, I don't think the RFD is being (accurately) represented, either. You may be getting a similar solution for different reasons.

How did you assess the causation of the claim in lines 346-347 ("…found to be a result of…")?

3. A hook echo does not *necessarily* confirm that a storm is a supercell, as is suggested in the paragraph beginning on line 236. Instead, focus on the rotation observed from radar velocity fields; the mesocyclone is the key defining feature of the supercell.

4. You seem to provide some details about storm initiation, evolution, and key ingredients in the summary and conclusion section that are not addressed earlier in the manuscript. Please discuss the evolution from a multicell cluster into a supercell and the importance of the "three ingredients" noted beginning at line 330 earlier in the manuscript, if you'd like to keep this section.

5. In several locations throughout the manuscript, there were brief, almost random mentions of climate change and the potential increase of extreme events in the future as a result. However, this line of thinking is never appreciably explored or elaborated upon; because of this, I think these mentions should be omitted.

6. In my opinion, the two paragraphs beginning at line 288 provide no additional insight. These appear to just be confirming the presence of an updraft through show-and-tell. If there is nothing more substantive to include, I would omit these paragraphs.

**Less Substantive Comments**
1. Why did you analyze such a small number of the available fields from ERA5? There are 137 vertical levels, and you only use three here. Could any insight be gained by interrogating additional fields/levels?

2. What was your reasoning for using the convergence of moisture flux at 800 mb rather than the lifted condensation level (LCL) as an assessment of low-level moisture?

3. For the paragraph beginning at line 206, it would be helpful to refer back to Fig. 1 to help illustrate the storm track.

4. A hodograph corresponding to the Irene sounding would be very helpful.

5. The 35 kt "lower mid-level jet" appears to be a bit stronger than analyzed in Fig. 2. Did this play a role?

6. Line 230: Why 573 hPa? Is there some specific importance of this level, or is it arbitrary?

7. How many observation sites are used to generate the plot in Fig. 8a? Are these sites reliable? The plot appears fairly noisy.

8. Likewise, how representative and reliable are the three sites used to create Fig. 8c?

**Minor/Grammatical Comments**

1. Line 18: "dependent" is misspelled

2. Line 29: Commas need before "such as" and after "fishing"

3. Line 46: "spectacular" is an odd word choice

4. Line 51: "results" should be singular ("result")

5. Lines 114-115: Suggest reworking, beginning with "Hourly data from a fifth generation European Centre…"

6. Line 120: Instead of "high", I suggested using "fine" to describe the grid spacing.

7. Line 128: Suggest omitting "spectacularly"

8. Lines 128-129: Rather than "an even stronger", I would say "a strong". The defining characteristic of a supercell is a strong, persistent rotating updraft.

9. Lines 156 and 166: Rather than "which", use "that" here.

10. Line 157: Omit comma after "damage".

11. Lines 160-162: I suggest reworking the sentence about storm damage. Rather than starting with "Visual impacts of the…", just note that "Damage from the storm included…", or something to that effect.

12. Line 173: I believe that "northward-extended" should read "northward-extending".

13. Line 174: "deepened" should maybe be "deepening", but it is difficult to determine given a lack of temporal context.

14. Line 180: "patterns" should be singular ("pattern")

15. Line 182: "advects" should be "advected"

16. Line 200: Add comma between "winds" and "which"

17. Line 212: Does the 1.5 km refer to the track length? This is unclear.

18. Line 249: I'm not sure what the word "near" is doing here.

19. Line 261, and others: Replace the comma before "however" with a semicolon, and place a comma after "however".

20. Line 348: Add a comma between "incorrect" and "possibly"

21. Generally, directions such as "southwestern" should not include a hyphen.

22. Fig. 4: This figure is a bit difficult to interpret/read because of poor image resolution.

---

## Referee Comment (RC1) · Anonymous Referee #1 · 14 Aug 2020

The authors of this manuscript present a case study of a tornadic supercell that affected parts of South Africa and led to extensive damage and injuries due to severe weather including a tornado that hit Vaal Marina in Gauteng Province. The study is relevant given the low number of tornado / supercell case studies in South Africa and can help to better understand conditions favorable for severe weather is this part of the country. I therefore recommend publication in Weather and Climate Dynamics. However, work is needed to increase the value of this paper since I have some concerns regarding some of the results presented in the conclusions, in particular the presented model forecasts.

Major comments

1) Results that need to be discussed in much greater depth to convince the reader

"SA1.5 underestimated mid-level vorticity due to a significant underestimation in low-level warm moisture advection and convergence." The lack of mid-level vorticity and low-level moisture flux convergence in the model field could be a consequence of the missed convection initiation. In other words: High mid-level vorticity and strong low-level moisture flux convergence would be a consequence of a supercell that develops in the model forecast. Since the model failed to initiate storms, it is also not able to increase the low-level moisture flux convergence and mid-level vorticity. See also lines 340-343: "The poor performance of SA1.5 seems to have emanated from a significant underestimation in low-level warm moisture advection and convergence (which act as dynamic lifting mechanisms), and as a result the mid-level vorticity being greatly underestimated, which led to the storm not being initialised."

"SA4.4 captured the supercell but underestimated its severity due to an underestimation in mid-level vorticity found to be one order of magnitude smaller than that of a typical mesocyclone." Since SA4.4 has limited resolution, wouldn't you expect that this already causes the vorticity of modelled supercells to be smaller?

"This was a result of underestimation in surface to mid-level wind shear and low-level horizontal mass and moisture flux convergence." Again, please discuss if the model resolution allows to directly indicate the effect of the developed convective storm on the presented parameters. Is it possible that you just analyse difference fields in the model that are affected by the initiated convective storm and therefore cannot be used to explain why a storm has formed or not? See also lines 283-287: "A relationship between MFC and vertical vorticity was also determined. It was found that SA4.4 predicted vorticity maxima at 500 hPa that is associated with MFC maxima at 800 hPa. MFC initiates first at 800 hPa followed by vorticity genesis at 500 hPa (Fig. 9(c)). This implies that the development of MFC in the low-levels could be a precursor of vertical vorticity initiation in the mid-levels. Since there is a positive relationship between low-level MFC and mid-level vorticity, it follows that the stronger the low-level MFC, the stronger the mid-level vorticity would be." It needs to be clear which of the presented model data are a consequence of convection initiation rather than ingredients for convective storms.

2) Ingredients-based methodology

I miss an analysis of the main ingredients for convective storms. These are low-level moisture, (mid-level) lapse rates, and lift. For low-level moisture and lift, I would highly appreciate a surface analysis including temperature, dewpoint, and wind barbs. The 15 UTC chart clearly indicates the presence of a dryline and shows the advection of moist air masses ahead of it. Are these drylines typical for that region or was this event special with respect to the intensity or location of the dryline? Two ascents have been operated at 9 UTC in the area of interest to indicate the presence of lapse rates (I recommend including FABL Bloemfontein Airport since it shows deep boundary layer mixing upstream although it is farther away). Please discuss if these soundings are representative since the launch time is many hours prior to the discussed event. Please check the sounding time. In the manuscript, the presented sounding is said to be launched at 12 UTC, but according to raw data it is launched at 9 UTC.

To discuss the magnitude of vertical wind shear, I also would include FABL Bloemfontein Airport sounding that has much greater mid-level winds upstream several hours prior to the event (45 knots at 500 hPa compared to 24 kts in the presented sounding).

Finally, dry mid-level air is not an ingredient for convective storms, e.g. lines 229-233: "The lower levels were moist with an average relative humidity of 91% between the surface and 625 hPa, while the mid-levels were dry with relative humidity averaging 26% between 573 hPa and 400 hPa. Therefore, the availability of low-level moisture, dry mid-levels, and vertical wind shearing provided favourable conditions for the possibility of dynamically and thermodynamically induced organised severe thunderstorms

(including supercell type) developing in the vicinity."

Further comments

1) Line 45: Single cells can produce severe weather, e.g., pulse storms.

2) Line 47: There can be also tornadoes not associated with intense convective storms (non-mesocyclonic tornadoes)

3) Line 79: Which criteria did you use to choose weather stations of interest? Please include a complete surface analysis that contains all data.

4) Figure 1: How did you produce the track of the cell? Radar site of the southern radar and the provincial borders are hardly visible in printed versions of the manuscript. Please include a reference scale.

5) Line 104-105: Please explain how the surface data were "regridded". What was the maximum distance between model grid points and surface data? Did you take terrain height differences into account? Was there an objective method to select proximity stations? Again, I would suggest a surface analysis.

6) Line 109: What was the distance of the radar sites from the supercell? You could solve this issue by including a reference scale in Figure 1.

7) Line 110-111: Please check the time between sounding launch and supercell event. The raw data file indicates that the soundings were launched three hours earlier. Please discuss if this has consequences for the classification as proximity sounding.

8) Lines 140-144: "The presence of boundary layer water vapour concentration is one of the most important factors for tornadogenesis (Markowski and Richardson, 2009). As a result, to analyse the significance of low-level moisture in the event considered, the convergence of moisture flux is computed at 800 hPa level. Moisture flux convergence (MFC) is a useful diagnostic tool as it combines the effects of moisture advection and convergence and can be computed at any atmospheric pressure level (Banacos

and Schultz, 2005)." MFC is not an ingredient for convective storms or tornadoes! Large MFC does not necessarily mean that moisture is high. Furthermore, at the observation sites shown in the paper, there is no moisture increase prior to the event. More discussion is needed to explain why you like to analyse MFC rather than moisture directly.

9) "This mid-level circulation patterns resulted in midatmospheric south-westerly winds over the interior of South Africa, which enhanced convective instability over the cen-traleast of the country as the dry mid-level air advects over the low-level warm and moist-air in the east." Dry mid-level air is not an ingredient for convective storms! There can be very dry air on top of very moist layers and still there is no instability. Please analyse the mid-level lapse rate field and look for regions where steep lapse rates overlap with rich low-level moisture to consider instability.

10) Figure 2: I would recommend using 300 hPa rather than 200 hPa.

11) Line 182-183: "A weak upper-air trough was also present over South Africa (Fig. 2(a))." If there is no further discussion on the upper level wind field, you may skip the 200 hPa chart. Otherwise, you may discuss in which way the upper trough / jet supported the development of a severe convective event.

12) Mesoanalysis: The presented data (Figures 3 and 4) are quite uncommon to anal-yse the mesoscale surface weather chart. A surface analysis would a good addition to these data.

13) Line 207: "After initiation, the storm initially propagated eastwards, then suddenly changed direction to north-east towards the Vaal Dam as it matured into a supercell thunderstorm through energy supply from a continual merger of several cells." This is very vague. Cell merger can also cause supercell decay due to the interaction of outflows. Why did this not happen in this case?

14) Mesoanalysis: I would re-organize this section. You start with surface observations,

jump to radar analysis, continue with the discussion of one surface observation that might be not representative, jump to the discussion of a sounding, jump back to radar analysis.

15) Line 260: It would be good to compare a surface chart directly to the winds in the lowest model level. The average regridded surface data does not allow for a detailed analysis. Furthermore, it is not explained how the averaging of the station data (due to which criteria are they chosen) is done.

---

## Author Comment (AC1) · 13 Oct 2020

"A dynamic and thermodynamic analysis of the 11 December 2017 tornadic supercell in the Highveld of South Africa"

**Response to reviewer #1**

**Reviewer:** The authors of this manuscript present a case study of a tornadic supercell that affected parts of South Africa and led to extensive damage and injuries due to severe weather including a tornado that hit Vaal Marina in Gauteng Province. The study is relevant given the low number of tornado / supercell case studies in South Africa and can help to better understand conditions favorable for severe weather is this part of the country. I therefore recommend publication in Weather and Climate Dynamics. However, work is needed to increase the value of this paper since I have some concerns regarding some of the results presented in the conclusions, in particular the presented model forecasts.

**Authors:** Thank you very much for your positive and detailed critical comments which have improved our manuscript. We provide a point-by-point response to all the major and minor comments below.

**Reviewer**: 1) Results that need to be discussed in much greater depth to convince the reader
The lack of mid-level vorticity and low-level moisture flux convergence in the model field could be a consequence of the missed convection initiation. In other words: High mid-level vorticity and strong low-level moisture flux convergence would be a consequence of a supercell that develops in the model forecast. Since the model failed to initiate storms, it is also not able to increase the low-level moisture flux convergence and mid-level vorticity.

**Authors**: Moisture flux convergence (MFC) has historically been used, from both observations and models, as a prognostic quantity for forecasting convective initiation (Banacos and Schultz, 2005). Hence, we concluded that a lack of MFC could be a precursor of, and not a consequence of, a lack of convective initiation or as in our case study, missed convective initiation. We attribute lack of storm during our analysis to lack of MFC. Perhaps there are more reasons, than just one, of why as noted by a recent study by Keat et al. (2019) which found that SA1.5 seems to consistently struggle with missed convection initiation over South Africa. They concluded that it might be due to physics set-up and the way sub-grid processes are parameterised in the model which might not be suitable for the region. In our revision we cite this study as one more possible reason why SA1.5 could be a case of missed convective initiation.

**Reviewer**: Since SA4.4 has limited resolution, wouldn't you expect that this

already causes the vorticity of modelled supercells to be smaller? Again, please discuss if the model resolution allows to directly indicate the effect of the developed convective storm on the presented parameters. Is it possible that you just analyse difference fields in the model that are affected by the initiated convective storm and therefore cannot be used to explain why a storm has formed or not?

**Authors**: Indeed Glickman (2000) indicated that the horizontal scale of the vortex is normally between 2 to 10km. This does indeed indicate that a grid length of 4.4km is too coarse (and in the grey zones really) to be able to resolve this process. We note that some previous studies (e.g. Weisman et al., 1997) have indicated that a grid spacing of 4 km may be sufficient to reproduce mesoconvective circulations and net momentum and heat transport of midlatitude type convective systems. We note also others who argue that models of grid spacing of 1 km or less are the ones adequate to represent dynamics and local processes responsible for triggering convection (see Roberts, 2008, and Bryan et al., 2003). Our revised manuscript will include a clear discussion on the effect of resolution on our results.

**Reviewer** 2) Ingredients-based methodology
I miss an analysis of the main ingredients for convective storms. These are low-level moisture, (mid-level) lapse rates, and lift. For low-level moisture and lift, I would highly appreciate a surface analysis including temperature, dewpoint, and wind barbs. The 15 UTC chart clearly indicates the presence of a dryline and shows the advection of moist air masses ahead of it. Are these drylines typical for that region or was this event special with respect to the intensity or location of the dryline? Two ascents have been operated at 9 UTC in the area of interest to indicate the presence of lapse rates (I recommend including FABL Bloemfontein Airport since it shows deep boundary layer mixing upstream although it is farther away). Please discuss if these soundings are representative since the launch time is many hours prior to the discussed event. Please check the sounding time. In the manuscript, the presented sounding is said to be launched at 12 UTC, but according to raw data it is launched at 9 UTC. To discuss the magnitude of vertical wind shear, I also would include FABL Bloemfontein Airport sounding that has much greater mid-level winds upstream several hours prior to the event (45 knots at 500 hPa compared to 24 kts in the presented sounding).

**Authors**: Thank you for this comment which made us realise that we missed to include the ingredients in a summarised format after our mesoanalysis. We include these in our revised manuscript. It was found that three ingredients were important in strengthening and maintaining this supercell: significant surface to midlevel vertical shear, an abundance of low-level warm moisture influx from the tropics and Mozambique Channel, and mid-level lapse rates. We also now include a surface chart analysis figure from the South African Weather Service which clearly indicates a dryline in the central interior of South Africa (especially the Free State Province). Regarding drylines, yes, they are typical especially in the central parts of South Africa during summer months. As recommended, we now include the FABL ascent which

clearly indicates mid-level winds upstream which we believe played a role in the formation of the supercell. We can also confirm that the sounding time was 12 UTC and not 9 UTC. This confirmation is from the South African Weather Service which is responsible for launching radiosondes. We also noted that raw data indicates 09 UTC, but the actual time was 12 UTC. We must say that this is a known problem and seems to be rather a technical error/bug.

Further comments
**Reviewer** 1) Line 45: Single cells can produce severe weather, e.g., pulse storms.

**Authors**: That is correct, however rare those cases are in South Africa, they do happen. We rectify this by noting that, however rare, single cells can also become severe as also noted by Ashley and Gilson (2009) who found that in some parts of the United States, lightning-related fatalities are most often produced by unorganized, pulse-style thunderstorms.

**Reviewer** 2) Line 47: There can be also tornadoes not associated with intense convective storms (non-mesocyclonic tornadoes)

**Authors**: Indeed there are non-supercellular tornadoes as also presented by Wakimoto and Wilson (1989).

**Reviewer** 3) Line 79: Which criteria did you use to choose weather stations of interest? Please include a complete surface analysis that contains all data.

**Authors**: We simply used stations that are closest to the storm track. The surface analysis is included in our revised manuscript which indicate all weather stations across South Africa.

**Reviewer** 4) Figure 1: How did you produce the track of the cell? Radar site of the southern radar and the provincial borders are hardly visible in printed versions of the manuscript. Please include a reference scale.

**Authors**: To produce the storm track we used the Thunderstorm Identification, Tracking, Analysis and Nowcasting (TITAN) and QGIS softwares to mark geographic information system (GPS) locations of the cell for each radar scan (for both radars in Irene and Ermelo, individual and merged for validation), from initiation to dissipation of the storm. The reference scale is now included in the figure.

**Reviewer** 5) Line 104-105: Please explain how the surface data were "regridded". What was the maximum distance between model grid points and surface data? Did you take terrain height differences into account? Was there an objective method to select proximity stations? Again, I would suggest a surface analysis.

**Authors:** The surface data was regridded using the Cressman objective analysis scheme which is described in detail by Cressman (1959). A Cressman objective analysis is performed on the station data to arrive at a gridded result that represents the station data. Model grid points are 4.4 km apart, while regridded surface data points are 25 km apart (hence it appears coarse). The regridding script does not take terrain height into account. However, observations are taken at 2 m (dewpoint and temperature) and 10 m (winds) above ground at each station. Proximity stations are simply taken to be those closest to the storm track, as indicated in Figure 1 and now in the surface chart analysis.

**Reviewer** 6) Line 109: What was the distance of the radar sites from the supercell? You could solve this issue by including a reference scale in Figure 1.

**Authors**: The distance varies (but within 150 km) depending on where the supercell was located at a particular time. We include a reference scale in our revised manuscript to resolve this.

**Reviewer** 7) Line 110-111: Please check the time between sounding launch and supercell event. The raw data file indicates that the soundings were launched three hours earlier. Please discuss if this has consequences for the classification as proximity sounding.

**Authors**: As already indicated in response to an earlier comment regarding radiosonde time, it was confirmed that the sounding was launched at 12 UTC, and not 09 UTC. This is therefore a good proximity sounding.

**Reviewer** 8) Lines 140-144: "The presence of boundary layer water vapour concentration is one of the most important factors for tornadogenesis (Markowski and Richardson, 2009). As a result, to analyse the significance of low-level moisture in the event considered, the convergence of moisture flux is computed at 800 hPa level. Moisture flux convergence (MFC) is a useful diagnostic tool as it combines the effects of moisture advection and convergence and can be computed at any atmospheric pressure level (Banacos and Schultz, 2005)." MFC is not an ingredient for convective storms or tornadoes! Large MFC does not necessarily mean that moisture is high. Furthermore, at the observation sites shown in the paper, there is no moisture increase prior to the event. More discussion is needed to explain why you like to analyse MFC rather than moisture directly.

**Authors**: We agree that MFC is not an ingredient for convective storms or tornadoes. Nor does high large MFC necessarily mean that moisture is high. We rather use MFC due to its combination of moisture advection and convergence. MFC has been used in the past as it typically occurs 3 hours prior to convective storms (see Hudson,1971; and Newman, 1971; Doswell, 1977; Negri and Vonder Haar, 1980; Waldstreicher, 1989). Not just moisture, but the concentration of moisture, has been shown to be

associated with areas of tornadogenesis. Perhaps the reference regarding boundary layer moisture concentration and tornadogenesis, followed by MFC causes misunderstandings. The use of tornadogenesis, or allusions thereof, seems to cause confusion as noted by Reviewer #2 as our focus in our model analysis is not primarily on tornadogenesis, rather we speak of tornadogenesis from observations data and not model data. Therefore, our revised manuscript removes any allusions of tornadogenesis in our model analysis. Lines 140-144 have been changed in our revised paper. Also, in our revised manuscript we integrate MFC from the surface to 600 hPa, and use that to define low-level following Ndarana et al. (2020).

**Reviewer** 9) "This mid-level circulation patterns resulted in mid-atmospheric south-westerly winds over the interior of South Africa, which enhanced convective instability over the central east of the country as the dry mid-level air advects over the low-level warm and moist-air in the east." Dry mid-level air is not an ingredient for convective storms! There can be very dry air on top of very moist layers and still there is no instability. Please analyse the mid-level lapse rate field and look for regions where steep lapse rates overlap with rich low-level moisture to consider instability.

**Authors**: Yes, we did not intend to communicate that a dry mid-level is an ingredient for convective storms, rather that the advection of dry mid-level air over low-level warm and moist-air enhances convective instability. Our phrasing is corrected in order to avoid confusion.

**Reviewer** 10) Figure 2: I would recommend using 300 hPa rather than 200 hPa.

**Authors**: We now use 300 hPa as suggested and we can see it has a relatively defined wave over the subcontinent than the 200 hPa.

**Reviewer** 11) Line 182-183: "A weak upper-air trough was also present over South Africa (Fig. 2(a))." If there is no further discussion on the upper level wind field, you may skip the 200 hPa chart. Otherwise, you may discuss in which way the upper trough / jet supported the development of a severe convective event.

**Authors**: Noted with thanks.

**Reviewer** 12) Mesoanalysis: The presented data (Figures 3 and 4) are quite uncommon to analyse the mesoscale surface weather chart. A surface analysis would a good addition to these data.

**Authors**: A surface chart is included for surface analysis in our revised manuscript.

**Reviewer** 13) Line 207: "After initiation, the storm initially propagated eastwards, then suddenly changed direction to north-east towards the Vaal Dam as it matured into a supercell thunderstorm through energy supply from a continual merger of several

cells." This is very vague. Cell merger can also cause supercell decay due to the interaction of outflows. Why did this not happen in this case?

**Authors**: Cell mergers in the southern African context are still not well understood. Typically, cell mergers are associated with storm intensification (not always), especially in development stages. We suspect storm directions and angle of "collision" played a role in intensifying the favoured storm which became a supercell. This requires further analysis which might be out of scope of our current study. What we know from our analysis is that there was a merger of several storms early on after storms initiated over the dry-line.

**Reviewer** 14) Mesoanalysis: I would re-organize this section. You start with surface observations, jump to radar analysis, continue with the discussion of one surface observation that might be not representative, jump to the discussion of a sounding, jump back to radar analysis.

**Authors**: Your suggestion is noted, and we try to re-organise our mesoanalysis so that its clear in our revised manuscript.

**Reviewer** 15) Line 260: It would be good to compare a surface chart directly to the winds in the lowest model level. The average regridded surface data does not allow for a detailed analysis. Furthermore, it is not explained how the averaging of the station data (due to which criteria are they chosen) is done.

**Authors**: This suggestion is helpful and greatly appreciated. The regridding method is already addressed in the previous comments.

**REFERENCE:**

Ashley, W. S., and C. W. Gilson, 2009: A Reassessment of U.S. Lightning Mortality. Bull. Amer. Meteor. Soc., 90, 1501–1518, https://doi.org/10.1175/2009BAMS2765.1.

Banacos, P.C., and Schultz, D.M.: The use of moisture flux convergence in forecasting convective initiation: Historical and operational perspectives, Wea. Forecasting, 20, 351 – 366. doi: https://doi.org/10.1175/WAF858.1, 2005.

Bryan, G.;Wyngaard, J.; Fritsch, J. Resolution Requirements for the Simulation of Deep Moist Convection. Mon. Weather Rev. 2003, 131. doi: https://doi.org/10.1175/1520-0493(2003)131%3C2394:RRFTSO%3E2.0.CO;2

Cressman, G. P., 1959: an operational objective analysis system. Mon. Wea. Rev., 87, 367–374, https://doi.org/10.1175/1520-0493(1959)087<0367:AOOAS>2.0.CO;2.

Doswell, C. A., III, 1977: Obtaining meteorologically significant surface divergence fields through the filtering property of objective analysis. Mon. Wea. Rev., 105, 885–892.

Hudson, H. R., 1971: On the relationship between horizontal moisture convergence and convective cloud formation. J. Appl. Meteor., 10, 755–762.

Keat, W.J., Stein, T.H.M., Phaduli, E., Landman, S., Becker, E., Bopape, M-J.M., Hanley, K.E., Lean, H.W., and Webster, S.: Convective initiation and storm life cycles in convection-permitting simulations of the Met Office Unified Model over South Africa, Quarterly Journal of the Royal Meteorological Society, 145, 1323-1336. doi: https://doi.org/10.1002/qj.3487, 2019

Ndarana, T, Mpati, S., Bopape, M-J.M., Engelbrecht, F., and Chikoore, H.: The flow and moisture fluxes associated with ridging South Atlantic Ocean anticyclones during the subtropical southern African summer, International Journal of Climatology, 1– 18. doi: https://doi.org/10.1002/joc.6745, 2020.

Negri, A. J., and T. H. Vonder Haar, 1980: Moisture convergence using satellite-derived wind fields: A severe local storm case study. Mon. Wea. Rev., 108, 1170–1182.

Newman, W. R., 1971: The relationship between horizontal moisture convergence and severe storm occurrences. M.S. thesis, School of Meteorology, University of Oklahoma, 54 pp. [Available from School of Meteorology, University of Oklahoma, 100 E. Boyd, Rm. 1310, Norman, OK 73019.]

Roberts, N. Assessing the spatial and temporal variation in the skill of precipitation forecasts from an NWP model. Meteorol. Appl. 2008, 15, 163–169. doi: https://doi.org/10.1002/met.57

Wakimoto, R. M., and J. W. Wilson, 1989: Non-supercell Tornadoes. Mon. Wea. Rev., 117, 1113–1140, https://doi.org/10.1175/1520-0493(1989)117<1113:NST>2.0.CO;2.

Weisman, M.; Skamarock, W.; Klemp, J. The Resolution Dependence of Explicitly Modeled Convective Systems. Mon. Weather Rev. 1997, 125. doi: https://doi.org/10.1175/1520-0493(1997)125%3C0527:TRDOEM%3E2.0.CO;2

---

## Author Comment (AC2) · 13 Oct 2020

**"A dynamic and thermodynamic analysis of the 11 December 2017 tornadic supercell in the Highveld of South Africa"**

**Response to reviewer #2**

**Reviewer:** Lekoloane et al. provide an interesting case study and a pair of model simulations associated with the 11 December 2017 tornadic supercell in the Highveld of South Africa. While this case would be of interest to the community, I have some serious reservations about the manuscript in its current form. In particular, there appear to be some fundamental misunderstandings—or at least omissions—related to the current knowledge base of supercell tornadogenesis. Additionally, I am concerned that the analyses of high-resolution simulations are too limited in scope in their current state and also reveal some misunderstandings. These issues could be alleviated with considerable revision and reanalysis, but I do not think the manuscript is suitable for publication at this time.

**Authors:** Thank you so much for taking the time to review and give feedback to our manuscript, which has led to an improved one. In the revised manuscript, we address all the major and minor comments the reviewer made, including those raised regarding tornadogenesis and model analysis. Below we make a point-by-point response to the reviewer's comments.

**Major Comments**

**Reviewer**: 1. Lines 57-59: Your discussion about near-surface vertical vorticity is short and potentially contains some misunderstandings. Tilting of environmental horizontal vorticity is the primary contributor to updraft rotation within a supercell, but it is not typically thought to contribute significantly to near-surface vertical vorticity. The other process that you mention could achieve near-surface vertical vorticity ("advection of vorticity from aloft") is vague, and I cannot be certain that what you mean here is actually consistent with current thinking. Please revise this section for clarity, and if necessary, review recent literature on the topic to ensure your introduction is consistent.

**Authors**: Thank you for the observation you make here, and we can see how our two sentences can cause confusion. Rather than go on mentioning more literature on tornadogenesis, which is a really complex and still less understood process (therefore potentially contradicting literature), we thought it best to remove the two sentences you mentioned here and other possible allusions to them, primarily because in this study our main focus is not on tornadogenesis itself, rather the supercell of which in our case was associated with a tornado. Hence, we mostly focused on the tornado in our observation analysis and only supercell in our model analysis in order to see if the configurations used in an operational setting at the South African Weather Service

(SAWS) were able to capture the supercell. This is because we acknowledge that the 1.5 km and 4.4 km models analysed in this study are too course to capture a tornado or even to do an analysis of tornadogenesis, but within a good range of capturing a supercell storm as found in a study by Weisman et al. (1997). Also, for your interest, our statements here were based on the referenced review paper by Markowski and Richardson (2009). Here's a direct quote from the paper: "By definition, tornadogenesis requires that large vertical vorticity arises at the ground. If preexisting vertical vorticity is negligible near the ground, then vorticity stretching near the ground is initially negligible and vertical vorticity first must arise either from the tilting of horizontal vorticity or from advection toward the surface from aloft." We understand this might be outdated given that more tornadogenesis studies have been done since then, and new theories developed.

**Reviewer**: 2a. Your method of evaluating model simulations is potentially flawed. SA1.5 very clearly lacks a supercell, as there was no precipitation present in the specific time frame. Thus, the use of midlevel vorticity was a strange choice to determine the presence of a supercell (or lack thereof). Additionally, you look at one hour here; could the higher-resolution simulation simply be delayed in time? What happened beyond 16z? This analysis seems to be somewhat cursory, and perhaps too superficial for a peer-reviewed manuscript.

**Authors**: In our revised manuscript we address this by including a figure of the 24-hour period accumulated precipitation spatial distribution for both model configurations. It does indicate that SA4.4 captured the storm while SA1.5 seems to be a case of missed convective initiation. We also include the Integrated Multi-satellitE Retrievals for Global Precipitation Measurement (GPM) (IMERG) and reanalysis (ERA5) precipitation fields which agree well with SA4.4 than SA1.5. We understand that reanalysis is not true observations, but is included to supplement satellite GPM observations. SA1.5 does not simulate any kind of precipitation throughout our analysis period. This was mentioned initially in the manuscript, however we did not show it. This seems to be a case of missed convective initiation, a similar problem found in a previous study by Keat et al. (2019) who also used the SA1.5. Keat et al. (2019) found that it might be due to physics set-up and the way sub-grid processes are parameterised which might not be suitable for South Africa. Since SA1.5 does not capture any storm in our domain during our 24-hour analysis period, we wanted to give possible reasons albeit with no proper reviewed-paper references, we fix this.

**Reviewer**: 2b. Your comment on lines 342-343, "…as a result of the mid-level vorticity being greatly underestimated, which led to the storm not being initialized" shows a fundamental misunderstanding of the source of modeled midlevel vorticity, or at least inconsistency with your analysis of SA4.4. The midlevel vorticity maxima/minima couplets in these simulations appear to be a consequence of the development of convection, not a cause. Please re-evaluate this statement.

**Authors**: Thanks for noting this, and we agree that the midlevel vorticity maxima/minima is a consequence of convection and not a cause. We remove this statement especially because SA1.5 failed to initiate any storms. We explain the probable cause by looking at MFC which is a good predictor of storm initiation. We also include reference of a study by Keat et al. (2019) which may explain why SA1.5 was a case of missed convective initiation.

**Reviewer**: 2c. Likewise, I have some concerns about your analysis and interpretations of SA4.4. As you noted, this simulation likely did not resolve the mesocyclone adequately, which is expected given the fairly coarse resolution (for a CAM). There are also some claims not entirely supported by the data provided, discussed below.

**Authors**: We agree that a grid length of 4.4km is not sufficient to resolve the mesocyclone and our study adds to other studies that have indicated that higher resolution is needed to resolve some processes in thunderstorms. A study by Weisman et al. (1997) indicated that a 4 km grid spacing model may be sufficient to reproduce mesoconvective circulations and net momentum and heat transport of midlatitude type convective systems. Other studies have however argued that models of grid spacing of 1 km or less are the ones adequate to represent dynamics and local processes responsible for triggering convection (see Roberts, 2008, and Bryan et al., 2003).

**Reviewer**: "…it follows that the stronger the low-level MFC, the stronger the mid-level vorticity would be." While this is likely just consequence of convergence below an updraft and the corresponding updraft, I'm not sure that the data support this claim in a more general sense. Do you have more evidence?

**Authors**: Yes we do have a figure which we now include. It indicates a positive correlation between low-level MFC and mid-level vorticity, and that low-level MFC occurs (or intensifies) about 2 to 3 hours prior to the development (or intensification) of mid-level vorticity. We now also note this as a result in our study not necessarily in a more general sense.

**Reviewer**: Lines 317-318: Given that SA4.4 doesn't adequately resolve the mesocyclone, I don't think you can expect it to accurately represent downdraft processes. RFDs tend to be considered dynamic features associated with low-level rotation within the storms. If the rotation isn't being resolved, I don't think the RFD is being (accurately) represented, either. You may be getting a similar solution for different reasons.

**Authors**: We agree that the model resolution may be too course to resolve the mesocyclone, and that RFDs are thought to be dynamic features associated with low-level rotation within supercells as also comprehensively reviewed by Markowski (2002) We have included in the manuscript the effect of resolution on our results.

**Reviewer**: How did you assess the causation of the claim in lines 346-347 ("…found to be a result of…")?

**Authors**: We compared observed vertical shear and modelled vertical shear. We also looked at simulated MFC, and now include reanalysis MFC to represent observations.

**Reviewer**: 3. A hook echo does not necessarily confirm that a storm is a supercell, as is suggested in the paragraph beginning on line 236. Instead, focus on the rotation observed from radar velocity fields; the mesocyclone is the key defining feature of the supercell.

**Authors**: Noted. Thanks for this great suggestion. In our revision we focus on the observed rotation from radar velocity fields.

**Reviewer**: 4. You seem to provide some details about storm initiation, evolution, and key ingredients in the summary and conclusion section that are not addressed earlier in the manuscript. Please discuss the evolution from a multicell cluster into a supercell and the importance of the "three ingredients" noted beginning at line 330 earlier in the manuscript, if you'd like to keep this section.

**Authors**: We realised we missed to include the ingredients in a summarised format after our mesoanalysis. We now include them. It was found that the three ingredients that were important in strengthening and maintaining this supercell are: significant surface to midlevel vertical shear, an abundance of low-level warm moisture influx from the tropics and Mozambique Channel, and mid-level lapse rates.

**Reviewer**: 5. In several locations throughout the manuscript, there were brief, almost random mentions of climate change and the potential increase of extreme events in the future as a result. However, this line of thinking is never appreciably explored or elaborated upon; because of this, I think these mentions should be omitted.

**Authors**: Thanks for your suggestion, these are removed.

**Reviewer**: 6. In my opinion, the two paragraphs beginning at line 288 provide no additional insight. These appear to just be confirming the presence of an updraft through show-and-tell. If there is nothing more substantive to include, I would omit these paragraphs.

**Authors**: Thank you very much for your thought. This is considered.

**Less Substantive Comments**

**Reviewer**: 1. Why did you analyze such a small number of the available fields from ERA5? There are 137 vertical levels, and you only use three here. Could any insight be gained by interrogating additional fields/levels?

**Authors**: Our original thinking was not to include more fields from reanalysis. But as explained in our previous responses, we now think reanalysis could help supplement some observations, and therefore we include more fields when performing the model evaluation.

**Reviewer**: 2. What was your reasoning for using the convergence of moisture flux at 800 mb rather than the lifted condensation level (LCL) as an assessment of low-level moisture?

**Authors**: We wanted to examine the lowest pressure level above the surface, which in our case was 800 hPa. For this case, moisture transport into South Africa is influenced by a South Atlantic anticyclone that extends eastward to induce an low level onshore flow into the eastern parts of the country from the South West Indian Ocean. This process is referred to as the ridging high. Therefore, the reason we did not use the LCL was due to the nature of the moisture transport by the ridging high pressure system, which is also influenced by the complex terrain in the eastern parts of the country.

In addition, our approach was to analyse the low-level moisture in general, rather than a specific level, we just didn't include other levels. To give more compelling evidence, in our revised manuscript we now integrate MFC from the surface to 600 hPa, instead of just 800 hPa. This follows Ndarana et al. (2020) who analysed the flow and moisture fluxes associated with ridging South Atlantic Ocean anticyclones during the subtropical southern African summer by integrating moisture fluxes from the surface to 600 hpa "because the flow over the Indian Ocean that is associated with ridging highs changes completely beyond this level to become sinusoidal and westerly, even as the flow in the lower levels is south-easterly." This insures that we analyse the entire low-level MFC. The new MFC equation reflecting the low-level convergence of moisture fluxes builds on equation 5 to become,

$$MFC^* = -\frac{1}{g} \int_{600}^{p_s} \nabla \cdot (q\bar{V}_h) dp$$

**Reviewer**: 3. For the paragraph beginning at line 206, it would be helpful to refer back to Fig. 1 to help illustrate the storm track.

**Authors**: Noted. Thanks!

**Reviewer**: 4. A hodograph corresponding to the Irene sounding would be very helpful.

**Authors**: Noted. In our revision we include it.

**Reviewer**: 5. The 35 kt "lower mid-level jet" appears to be a bit stronger than analyzed in Fig. 2. Did this play a role?

**Authors**: Perhaps, but it would be difficult to confidently say so in our study as it seems to be a common feature and would therefore require a climatological baseline which is currently not available.

**Reviewer**: 6. Line 230: Why 573 hPa? Is there some specific importance of this level, or is it arbitrary?

**Authors**: This is arbitrary.

**Reviewer**: 7. How many observation sites are used to generate the plot in Fig. 8a? Are these sites reliable? The plot appears fairly noisy.

**Authors**: The figure is generated from 228 automatic weather stations across South Africa. The surface data was regridded using the Cressman objective analysis scheme which is described in detail by Cressman (1959) . A Cressman objective analysis is performed on the station data to arrive at a gridded result that represents the station data. Model grid points are 4.4 km apart, while regridded surface data points are 25 km apart. Hence it appears coarse.

**Reviewer**: 8. Likewise, how representative and reliable are the three sites used to create Fig. 8c?

**Authors**: Unfortunately these are those closest to the storm track. We do have a figure indicating that there is consistency between different station.

**REFERENCE:**

Bryan, G.;Wyngaard, J.; Fritsch, J. Resolution Requirements for the Simulation of Deep Moist Convection. Mon. Weather Rev. 2003, 131. doi: https://doi.org/10.1175/1520-0493(2003)131%3C2394:RRFTSO%3E2.0.CO;2

Cressman, G. P., 1959: an operational objective analysis system. Mon. Wea. Rev., 87, 367–374, https://doi.org/10.1175/1520-0493(1959)087<0367:AOOAS>2.0.CO;2.

Keat, W.J., Stein, T.H.M., Phaduli, E., Landman, S., Becker, E., Bopape, M-J.M., Hanley, K.E., Lean, H.W., and Webster, S.: Convective initiation and storm life

cycles in convection-permitting simulations of the Met Office Unified Model over South Africa, Quarterly Journal of the Royal Meteorological Society, 145, 1323-1336. doi: https://doi.org/10.1002/qj.3487, 2019

Markowski, P. M., 2002: Hook Echoes and Rear-Flank Downdrafts: A Review. Mon. Wea. Rev., 130, 852–876, https://doi.org/10.1175/1520-0493(2002)130<0852:HEARFD>2.0.CO;2.

Markowski, P.M., and Richardson, Y.P.: Tornadogenesis: Our current understanding, forecasting considerations, and questions to guide future research, Atmospheric Research, 93, 3-10. doi: https://doi.org/10.1016/j.atmosres.2008.09.015, 2009

Ndarana, T, Mpati, S., Bopape, M-J.M., Engelbrecht, F., and Chikoore, H.: The flow and moisture fluxes associated with ridging South Atlantic Ocean anticyclones during the subtropical southern African summer, International Journal of Climatology, 1– 18. doi: https://doi.org/10.1002/joc.6745, 2020.

Roberts, N. Assessing the spatial and temporal variation in the skill of precipitation forecasts from an NWP model. Meteorol. Appl. 2008, 15, 163–169. doi: https://doi.org/10.1002/met.57

Weisman, M.; Skamarock, W.; Klemp, J. The Resolution Dependence of Explicitly Modeled Convective Systems. Mon. Weather Rev. 1997, 125. doi: https://doi.org/10.1175/1520-0493(1997)125%3C0527:TRDOEM%3E2.0.CO;2

---

## Referee Report (RR1)

**Second Review of WCD-2020-34**

**Authors:** Lesetja E. Lekoloane et al.
**Title:** "A dynamic and thermodynamic analysis of the 11 December 2017 tornado supercell in the Highveld of South Africa"
**Recommendation:** Acceptable pending minor revisions

**Overview**
The second iteration of Lekoloane et al.'s manuscript is much more focused than the original submission, improving its readability and suitability. The manuscript still requires a considerable amount of mostly minor revisions and grammatical corrections, which are noted below.

I also have some major comments that I think need to be addressed. In particular, I do not feel that attribution to stated environmental ingredients is clearly demonstrated in the manuscript, though these ingredients did likely play a role in the storm's development and evolution. Also, I am not confident that modeled vorticity would be any more useful or accurate than the wind fields themselves, from which vorticity is derived. Addressing these concerns would mostly require some additions or changes to the narrative.

Following revisions, I am happy to review the paper again or allow the editor to make a final decision on the manuscript.

**Major Comments**
1. The strengthening and maintenance of the supercell is not clearly attributed to the three ingredients mentioned (vertical shear, low-level warm/moist flux, and dry midlevels). These all play a role in producing a suitable environment for a supercell to develop and thrive, but it is not clear how they specifically strengthened and maintained the supercell here. The easiest way to alleviate this issue is to adjust your wording somewhat, unless you can justify your statement with more rigorous cause-and-effect testing.

2. Beginning of section 2.2.4: It is not clear to me why vorticity would be suitable here when it is derived from the components that you note cannot be reliably used. You should provide some reasoning for why you think vorticity is not subject to the same resolution concerns.

**Less Substantive Comments**
1. Line 35: What "development scale" are you referring to? More detail would be useful here, as I assume many readers will be unfamiliar with this scale.

2. Line 48: I would argue that any thunderstorm that produces a tornado is a "severe" storm. I recommend rewording this sentence as, "It should be noted that some multicell thunderstorms can also produce tornadoes (including non-mesocyclonic tornadoes), and…"

3. Line 68: Omit or separate "including several animals", as these should not be included in the human injuries.

4. Line 228: "through energy supply" is vague and speculative. I would omit this.

5. Lines 240-241: What is the significance of the convergence noted here? Explain why it is worth mentioning.

6. Line 260: Specifically, the advection of dry air atop warm, moist air builds "potential" instability.

7. Line 316: "…which is an indicator that SA4.4 did not capture the mesocyclone of the VAM storm." While this is likely true, I wonder if the magnitude is low because of the chosen pressure level (500 mb). How do 700-mb and 850-mb vertical vorticity compare? These levels may more accurately depict the midlevel or low-level mesocyclone.

8. Lines 319-321: This claim is speculative. Though it may be true, do you have any support for it?

9. Lines 385-386: Rather than referring to each component as an "underestimation", I suggest noting the "poor resolution" of the given features due to 4.4-km grid spacing.

10. Fig. 3: The resolution of this figure is poor, making it difficult to read the surface observations and identify the noted features. If you cannot achieve higher resolution of the hand analysis, I recommend including a figure of the surface observations and trying to digitally recreate the hand analysis, or at least annotate the key features.

11. Fig. 5: These are nice plots, but I think they would be more readable in two rows and four columns.

12. Fig. 6: The surface point is clearly erroneous in the FAIR sounding. Can you remove it when plotting?

13. Fig. 7: Higher resolution images would be very helpful here.

14. Fig. 8: This is a great addition and succinctly addresses some of my prior comments. Thanks for including it!

15. Fig. 9: The surface winds here seem quite high and seem to be at odds with the prior surface map, though it is a little difficult to tell based on the resolution of Fig. 3. For example, in Fig. 3, I do not see any observations of 30 kt or higher. This could be a difference in timing (12z vs. 15z), but I also want to ensure that the data in Fig. 9 are accurate.

16. Fig. 12: These plots are actually showing divergence (since negative values are associated with convergence). Please reflect this in your titles.

17. Fig. 12: It may also be helpful to show some measure of surface temperature, theta, or theta-E to delineate cold pools.

**Minor/Grammatical Comments**
1. Line 10: "that of 1.5 km grid spacing (SA1.5) version of it" should read "the 1.5-km grid spacing version."

2. Line 11 and throughout: "however" should be preceded by a semicolon and followed by a comma; e.g., "SA4.4 captured the supercell; however, the mid-level…"

3. Line 20: Comma needed between "society" and "including"

4. Line 24: "occurred" misspelled

5. Line 24: Change "which" to "and"

6. Line 25: Based on the preceding sentences, I would recommend starting the sentence with "There" instead of "Therefore" (i.e.,, "There is growing evidence…")

7. Line 25: Moreover, change the following wording of the sentence: "…growing evidence suggesting that these extreme events…"

8. Line 26: Omit comma after "globe"

9. Line 29: Change sentence to read "…the most vulnerable communities are those in developing countries."

10. Line 30: "depended" should be "dependent"

11. Line 30: Comma needed after "fishing"

12. Line 35: "lacking" should be "lagging"

13. Line 35: Comma needed between "stressors" and "including"

14. Line 42: Comma needed before "mostly"

15. Line 44: Change "which" to "that"

16. Line 44: Comma needed before "including"

17. Line 46: Change en dash to semicolon

18. Line 64: Change mid-level to mid levels

19. Line 64: "rotation at the ground" should be "the development of rotation at the ground"

20. Line 77: Comma needed between "understanding" and "the"

21. Line 79: Omit "models of"

22. Lines 79-80: Change "are the ones adequate" to "is necessary"

23. Line 81: Change "a model run" to "model runs" and "a grid length" to "grid lengths"

24. Line 88: Omit commas around "in this study"

25. Line 112 and throughout: Comma needed before "which"

26. Line 116: Omit comma after "Africa"

27. Line 129: "upstreams" should be "upstream"

28. Line 136: Be consistent with hyphens in noting pressure levels

29. Lines 136-137: Font size changes, and there is a hanging slash at the end of the sentence.

30. Line 145: "represents" should be "represent"

31. Line 162: Omit comma after "hPa"

32. Line 176: Change "which" to "that"

33. Line 177: Omit comma after "damage"

34. Line 201: Change "advects" to "advected"

35. Line 203: Omit hyphen after "western"

36. Line 233: Change comma after "hail" to an "and"

37. Line 264: Remove hyphen in "low levels"

38. Line 271: Semicolon should be a comma or omitted

39. Line 276: Use "prior to" instead of your second "before" to prevent redundancy

40. Line 276: Change "near storm" to "the near-storm"

41. Line 278: Add comma after "Machadodorp"

42. Line 281: Change "twenty-four hour" to "24-hour"

43. Line 284: Include "a" between "ERA5 shows" and "large"

44. Line 295 and throughout: Add hyphen in "10-m"

45. Line 329: Omit comma before "reveals"

46. Line 330 and throughout: I've always heard and seen "poleward" rather than "polarward"

47. Line 339: Omit hyphen in "moist air"

48. Line 362: Omit comma after "underestimated"; could also add a comma before "possibly"

49. Line 378: Unneeded commas around "and therefore"

50. Line 380: Add a period after "et al"

51. Line 388: At least one word is missing between "was" and "which"

52. Huffman et al. reference is out of alphabetical order.

---

## Referee Report (RR2)

**Comments on "A dynamic and thermodynamic analysis of the 11 December 2017 tornadic supercell in the Highveld of South Africa"**

The new version of the publication "A dynamic and thermodynamic analysis of the 11 December 2017 tornadic supercell in the Highveld of South Africa" has improved. The authors included additional material as suggested and addressed some of the comments raised during the review.

However, the answers to some comments are not convincing yet. This applies in particular to the major comments. Lines 375 to 389 contain essentially the same passages of text as the original submission, although the authors have inserted some relativizing statements in the current manuscript (major comment #1). The same goes for major comment #2. Here, an analysis of 2m temperature, dew point, and wind would improve the publication. The analysis provided in the recent resubmission does not help to analyse the situation with respect to the mesoscale situation. A deeper discussion on the distribution of environmental lapse rates would help to understand how necessary ingredients of convective storms came together is this situation. The comment on the importance of dry air at mid levels has not yet been addressed by the authors yet.

Finally, the structure of the mesoanalysis has not improved significantly. It is still difficult to follow the section since there are jumps between different parts of the analysis.

---

## Author Response (AR2)

**"A dynamic and thermodynamic analysis of the 11 December 2017 tornadic supercell in the Highveld of South Africa"**

**Response to referee report #1**

**Major Comments**

**Reviewer:** 1. The strengthening and maintenance of the supercell is not clearly attributed to the three ingredients mentioned (vertical shear, low-level warm/moist flux, and dry midlevels). These all play a role in producing a suitable environment for a supercell to develop and thrive, but it is not clear how they specifically strengthened and maintained the supercell here. The easiest way to alleviate this issue is to adjust your wording somewhat, unless you can justify your statement with more rigorous cause-and-effect testing.

**Response:** Thank you for pointing this out. We have adjusted our wording by including prevailing conditions in the eastern parts of the country (new figures showing low-level moisture, midlevel environmental lapse rates, and winds), and therefore indicated that they most likely played a role in ensuring that the supercell storm is strengthened and maintained throughout its lifetime.

**Reviewer:** 2. Beginning of section 2.2.4: It is not clear to me why vorticity would be suitable here when it is derived from the components that you note cannot be reliably used. You should provide some reasoning for why you think vorticity is not subject to the same resolution concerns.

**Response:** To resolve this, we reference Stevens and Crum (2003) and indicate that an added advantage is that it includes most of the flow and is also much easier to use for rotation analysis compared to only using wind vectors.

**Less Substantive Comments**

**Reviewer:** 1. Line 35: What "development scale" are you referring to? More detail would be useful here, as I assume many readers will be unfamiliar with this scale.

**Response:** Thank you for this suggestion. We now reference a short paper by Carley and Bustelo (1986) which briefly explains the concept of social indicators for the reader unfamiliar with them.

**Reviewer:** 2. Line 48: I would argue that any thunderstorm that produces a tornado is a "severe" storm. I recommend rewording this sentence as, "It should be noted that some multicell thunderstorms can also produce tornadoes (including non-mesocyclonic tornadoes), and..."

**Response:** This has been noted in the revised manuscript.

**Reviewer:** 3. Line 68: Omit or separate "including several animals", as these should not be included in the human injuries.

**Response:** This has been omitted.

**Reviewer:** 4. Line 228: "through energy supply" is vague and speculative. I would omit this.

**Response:** This has been omitted as suggested.

**Reviewer:** 5. Lines 240-241: What is the significance of the convergence noted here? Explain why it is worth mentioning.

**Response:** This is resolved by indicating that It is important to note this because, studies conducted by Seko et al. (2015) and Yokota et al. (2016), revealed that low-level water vapour and convergence near the storm are important factors for low-level mesocyclogenesis, which is a process important in supercell tornadogenesis.

**Reviewer:** 6. Line 260: Specifically, the advection of dry air atop warm, moist air builds "potential" instability.

**Response:** This is noted thanks.

**Reviewer:** 7. Line 316: "…which is an indicator that SA4.4 did not capture the mesocyclone of the VAM storm." While this is likely true, I wonder if the magnitude is low because of the chosen pressure level (500 mb). How do 700-mb and 850-mb vertical vorticity compare? These levels may more accurately depict the midlevel or low-level mesocyclone.

**Response:** Although we do realise that the suggested levels may be true generally speaking, during our exploratory stage we did analyse these levels and found that they are well represented by the 500hPa in our case study.

**Reviewer:** 8. Lines 319-321: This claim is speculative. Though it may be true, do you have any support for it?

**Response:** Thank you for pointing this out. The co-editor also noted this, and therefore we have dropped this claim.

**Reviewer:** 9. Lines 385-386: Rather than referring to each component as an "underestimation", I suggest noting the "poor resolution" of the given features due to 4.4-km grid spacing.

**Response:** The poor resolution of the 4.4-km grid spacing model has now been noted in the revised manuscript.

**Reviewer:** 10. Fig. 3: The resolution of this figure is poor, making it difficult to read the surface observations and identify the noted features. If you cannot achieve higher resolution of the hand analysis, I recommend including a figure of the surface observations and trying to digitally recreate the hand analysis, or at least annotate the key features.

**Response:** This figure has now been replaced with a much clearer one.

**Reviewer:** 11. Fig. 5: These are nice plots, but I think they would be more readable in two rows and four columns.

**Response:** Thank you for this suggestions. We have now changed them to two rows and four columns.

**Reviewer:** 12. Fig. 6: The surface point is clearly erroneous in the FAIR sounding. Can you remove it when plotting?

**Response:** Although we could not change this, the "hypothetical" parcel trajectory indicated by the black line is based on data masking this surface erroneous data as can be seen on the figure.

**Reviewer:** 13. Fig. 7: Higher resolution images would be very helpful here.

**Response:** Unfortunately this is caused by the radar resolution. When we apply smoothing schemes it affects some of the important features, hence this is probably the best possible plot without affecting the "true state" of the observation.

**Reviewer:** 14. Fig. 8: This is a great addition and succinctly addresses some of my prior comments. Thanks for including it!

**Response:** Thank you for suggesting that we add it!

**Reviewer:** 15. Fig. 9: The surface winds here seem quite high and seem to be at odds with the prior surface map, though it is a little difficult to tell based on the resolution of Fig. 3. For example, in Fig. 3, I do not see any observations of 30 kt or higher. This could be a difference in timing (12z vs. 15z), but I also want to ensure that the data in Fig. 9 are accurate.

**Response:** This seems to be due to the difference in timing. As can be seen in the figure indicating wind roses, Kroonstad, Klerksdorp and Ermelo do records winds of over 30kts and at times over 40kts.

**Reviewer:** 17. Fig. 12: It may also be helpful to show some measure of surface temperature, theta, or theta-E to delineate cold pools..

**Response:** Surface temperature is now included to capture this suggestion.

**Reference**

Stevens, D.E., and Crum, F.X.: Meteorology, Dynamic (Troposphere). Encyclopedia of Physical Science and Technology [Meyers, R.A. (ed)], 3rd Ed., Academic Press, San Diego, 629–659. doi: https://doi.org/10.1016/B0-12-227410-5/00436-1, 2003.

Carley, M, and Bustelo, E.: Social indicators and development, Project Appraisal, 1, 266–268. doi: https://doi.org/10.1080/02688867.1986.9726580, 1986.

Seko, H., Kunii, M., Yokota, S., Tsuyuki, T., and Miyoshi, T.: Ensemble experiments using a nested LETKF system to reproduce intense vortices associated with tornadoes of 6 May 2012 in Japan, Prog. Earth Planet. Sci., 2, 42. doi: https://doi.org/10.1186/s40645-015-0072-3, 2015.

Yokota, S., Seko, H., Kunii, M., Yamauchi, H., and Niino, H.: The tornadic supercell on the Kanto Plain on 6 May 2012: Polarimetric radar and surface data assimilation with EnKF and ensemble-based sensitivity analysis, Mon. Wea. Rev., 144, 3133–3157. doi: https://doi.org/10.1175/MWR-D-15-0365.1, 2016.

**"A dynamic and thermodynamic analysis of the 11 December 2017 tornadic supercell in the Highveld of South Africa"**

**Response to referee report #2**

**Reviewer**: The new version of the publication "A dynamic and thermodynamic analysis of the 11 December 2017 tornadic supercell in the Highveld of South Africa" has improved. The authors included additional material as suggested and addressed some of the comments raised during the review.

**Authors**: We would like to thank the reviewer for making time to review this manuscript. The review helped improve our manuscript. We provide a point-by-point response to all the major comments below.

**Reviewer**: Lines 375 to 389 contain essentially the same passages of text as the original submission, although the authors have inserted some relativizing statements in the current manuscript

**Authors**: Thank you for pointing this out. To fix this we have written this section in such a way that is much more clearer and have added more supporting references.

**Reviewer**: Here, an analysis of 2m temperature, dew point, and wind would improve the publication. The analysis provided in the recent resubmission does not help to analyse the situation with respect to the mesoscale situation. A deeper discussion on the distribution of environmental lapse rates would help to understand how necessary ingredients of convective storms came together is this situation. The comment on the importance of dry air at mid levels has not yet been addressed by the authors yet.

**Authors**: This is addressed this by including surface temperature, dewpoint and wind analysis plots, from the surface chart and reanalysis. We have also included a detailed discussion of the environment lapse rate (ELR) analysis alongside other convective storms ingredients already noted. ELR analysis also helps resolves the question surrounding the importance of dry mid-level air.